# Coherent heteroepitaxial growth of I-III-VI$_2$ Ag(In,Ga)S$_2$ colloidal nanocrystals with near-unity quantum yield for use in luminescent solar concentrators

Hak June Lee [1,2,8], Seongbin Im [1,8], Dongju Jung [1,8], Kyuri Kim[1], Jong Ah Chae[1], Jaemin Lim [1], Jeong Woo Park [1], Doyoon Shin [1], Kookheon Char[2], Byeong Guk Jeong[3], Ji-Sang Park[1], Euyheon Hwang [1], Doh C. Lee [4], Young-Shin Park [5], Hyung-Jun Song [6] ✉, Jun Hyuk Chang [7] ✉ & Wan Ki Bae [1] ✉

Colloidal Ag(In,Ga)S$_2$ nanocrystals (AIGS NCs) with the band gap tunability by their size and composition within visible range have garnered surging interest. High absorption cross-section and narrow emission linewidth of AIGS NCs make them ideally suited to address the challenges of Cd-free NCs in wide-ranging photonic applications. However, AIGS NCs have shown relatively underwhelming photoluminescence quantum yield (PL QY) to date, primarily because coherent heteroepitaxy has not been realized. Here, we report the heteroepitaxy for AIGS-AgGaS$_2$ (AIGS-AGS) core-shell NCs bearing near-unity PL QYs in almost full visible range (460 to 620 nm) and enhanced photo-chemical stability. Key to the successful growth of AIGS-AGS NCs is the use of the Ag-S-Ga(OA)$_2$ complex, which complements the reactivities among cations for both homogeneous AIGS cores in various compositions and uniform AGS shell growth. The heteroepitaxy between AIGS and AGS results in the Type I heterojunction that effectively confines charge carriers within the emissive core without optically active interfacial defects. AIGS-AGS NCs show higher extinction coefficient and narrower spectral linewidth compared to state-of-the-art heavy metal-free NCs, prompting their immediate use in practicable applications including displays and luminescent solar concentrators (LSCs).

Colloidal semiconductor NCs are solution processable nano-emitters that emanate size-dependent tunable energies of photons with a narrow linewidth[1–3]. High-precision size control of NCs permits to fine-tune the emission wavelength in a nanometer scale[4–6]. In addition, defect-free heteroepitaxy in a core–shell geometry allows stable and efficient radiation of NCs under various circumstances[7–10]. These advances in chemistry accelerate the use of NCs in a range of photonic applications including displays[11–14], lasers[15,16], imagings[17,18], and energy conversion systems[19–22].

Demand for practical use in everyday light-emitting applications drives to expand the materials envelope for core–shell NCs toward semiconductors free from heavy metal elements[23–31]. Among potential candidates, I–III–VI$_2$ alloyed compounds made of Ag, (In,Ga) and S have been of particular interest, as they promise efficient light absorption, wide-ranging bandgap tunability from visible to near-IR and a narrow emission linewidth[32–38]. Nevertheless, only marginal success has been reported in the synthesis and structural engineering

of AIGS cores. Mostly, an effective way of passivating the surface without rendering unwanted defects, which is the key to efficiency and stability, has yet to be developed.

Herein, we devise a chemical route for high-quality AIGS-AGS core−shell NCs. Specifically, a molecular precursor containing Ag, Ga, and S (Ag−S−Ga(OA)$_2$) is deployed for homogeneous AIGS nucleation of various compositions and uniform AGS shell growth, enabling us to realize AIGS-AGS NCs displaying color pure emissions in a wide visible region with near-unity PL QYs. We conduct spectroscopic analysis to identify the impact of AGS heteroepitaxy on the photophysical and chemical properties of AIGS cores and finally discuss the competitive advantages of AIGS-AGS NCs in photonic applications.

## Results

### Coherent heteroepitaxial growth of AIGS-AGS NCs

The up-to-date chemistry still faces hurdles in attaining homogeneity in AIGS alloyed core synthesis and passivating the surface trap states, as seen from their mediocre optical characteristics (see Table 1 and Supplementary Fig. 1)[34–37,39]. The broad emission linewidth relates to the size and/or composition inhomogeneities among AIGS alloyed cores, which are attributed to substantial differences in reactivity among cation precursors, specifically the low reactivity of Ga precursors compared to their cation counterparts (Ag or In precursors). The reactivity of In precursors (In(OA)$_3$) far exceeds that of Ga precursors (Ga(OA)$_3$), as predicted from the substantial bond-dissociation energy difference between Ga$^{3+}$ versus In$^{3+}$ to carboxylate ligands (OA) (Supplementary Table 1). In addition, Ag precursor is prone to be reduced to precipitate in a form of silver metal (Ag) at an elevated temperature, at which Ga precursor is activated to participate in the chemical reaction (see Supplementary Fig. 2). Therefore, the conventional hot injection method, i.e., the injection of S precursor into the mixed solution of Ag, In and Ga precursors, is not applicable for homogenous AIGS core synthesis unless the reactivities of cation precursors are complemented.

At the same time, the chemistry for optically active defect-free heteroepitaxy is necessary to boost the optical performance of AIGS cores. Previously, zincblende ZnS[34,35] or amorphous GaS$_x$[36,37] have been suggested, but the structural difference between AIGS core and these shell materials has yielded interfacial defects that deteriorate the spectral purity and luminescence efficiency of resulting NCs. In this respect, AGS is ideal for the shell materials not only because its crystal structure is same to AIGS with a manageable lattice mismatch (i.e., the mismatch between AIGS ($X = 0.5$) versus AGS = 1.2% in $a$ axis and 3.9% in $c$ axis) but also because it constructs straddling energy gap (Type I

heterojunction) that facilitates efficient recombination of charge carriers in the AIGS phase. Obviously, for the AGS shell growth, the reactivities of cation precursors should be managed to avoid unwanted side reactions (homogeneous nucleation of Ag$_2$S nanoparticles (NPs) or precipitation of Ag).

The above consideration imposes a chemical means to balance the reactivities among cation precursors for achieving AIGS-AGS core−shell NCs. In this study, we devise a molecular precursor containing Ag, Ga, and S (Ag−S−Ga(OA)$_2$) for both AIGS core synthesis and AGS shell growth (Fig. 1a). Ag−S−Ga(OA)$_2$ stock solution is prepared in following reaction steps (see "Methods" for detailed procedures). (i) Ag$_2$S NPs are formed to avoid Ag reduction at an elevated temperature. (ii) Ag$_2$S NPs react with in-situ generated H$_2$S from the mixture of S and alkylamine at an elevated temperature ($T \geq 130\,°C$) and dissolve into a form of AgSH. (iii) AgSH reacts with Ga(OA)$_3$ to form Ag−S−Ga(OA)$_2$.

The role of in situ generated H$_2$S in preparation of Ag−S−Ga(OA)$_2$ is validated by the comparative experiments with versus without degassing for H$_2$S removal (Supplementary Figs. 3 and 4). Specifically, we prepare S-OAm stock solutions degassed at two different temperatures, i.e., room temperature (RT) and 130 °C (at which, the in situ generated H$_2$S gas is degassed and collected in cold trap connected to the Schlenk line as yellowish-brown color) (Supplementary Figs. 3 and 4). These S-OAm stock solutions (degassed at RT versus 130 °C) are separately added into the reaction flasks containing Ag$_2$S NPs and the reaction temperature is gradually elevated to 210 °C. Ag$_2$S NPs mixed with S-OAm degassed at RT are decomposed into a form of AgSH (Ag$_2$S NPs + H$_2$S → AgSH, molar ratio of Ag:S = 1:1 is confirmed by inductively coupled plasma atomic emission spectroscopy (ICP-AES)), as seen in the gradual decrease in absorbance of Ag$_2$S NPs and transmission electron microscope (TEM) analysis (Supplementary Figs. 3a and 5). In the presence of Ga(OA)$_3$, AgSH simultaneously reacts with Ga(OA)$_3$ to yield Ag−S−Ga(OA)$_2$ complex. By contrast, Ag$_2$S NPs mixed with S-OAm degassed at 130 °C remain unchanged until 170 °C and decompose into elemental Ag and S at 210 °C (Supplementary Fig. 3b). In this case, the present Ga(OA)$_3$ does not participate in the reaction (Supplementary Fig. 4). Above results clearly show that in situ generated H$_2$S is the key element to produce Ag−S−Ga(OA)$_2$ complex.

The formation of Ag−S−Ga(OA)$_2$ is confirmed with matrix-assisted laser desorption ionization-time of flight (MALDI-TOF) mass spectrometry and optical characterization (Fig. 1b and Supplementary Figs. 5 and 6). Ag−S−Ga(OA)$_2$ stock solution is kept at the RT before it was used for AIGS core synthesis and AGS shell growth. The injection of In precursor (In(OA)$_3$) into Ag−S−Ga(OA)$_2$ stock solution at an elevated temperature ($T = 210\,°C$) bursts AIGS nucleation and subsequent growth. The addition of Ag−S−Ga(OA)$_2$ stock solution in the AIGS core containing solution allows to grow AGS shell on top of AIGS core. High-resolution TEM (HR-TEM), scanning TEM-high angle annular dark field imaging (STEM-HAADF), and energy dispersive spectroscopy (EDS) analysis verify that resulting NCs are indeed heterostructured in AIGS-AGS core−shell geometry (Fig. 1c−e). It is noted that excess amount of S is deployed throughout the core−shell growth as the S-rich reaction condition guarantees higher PL QY and suppressed trap emission of resulting AIGS-AGS NCs (Supplementary Fig. 7). Reaction procedures for precursor preparations and core−shell growth are detailed in "Methods".

### Structural and optical characteristics of AIGS-AGS NCs

AIGS core and AGS shell build Type I heterojunction, in which both electron and hole wavefunctions are confined within AIGS core (Fig. 2a). In such potential profile, large versatility in controlling the chemical composition of AIGS core and the dimension of AGS shell provide AIGS-AGS NCs with brightness and stability as well as with expansive emission tunability. The reaction scheme enables uniform AGS shell growth with varying thicknesses (Fig. 2b and Supplementary

**Table 1 | Optical properties of heavy metal-free NCs in the literature**

|  | PL maximum (nm) | FWHM (nm) | PL QY (%) | Ref. |
|---|---|---|---|---|
| InP-ZnSeS | 528 | 36 | 95 | 25 |
|  | 525 | 38 | 88 | 26 |
|  | 531 | 34 | 68 | 26 |
|  | 518 | 42 | 78 | 27 |
|  | 527 | 36 | 97 | 28 |
| ZnSeTe–ZnSeS | 524 | 41 | 95 | 29 |
|  | 520 | 45 | 80 | 30 |
|  | 500 | 43 | 88 | 31 |
| AIGS-ZnS | 544 | 137 | 68 | 34 |
|  | 618 | 100 | 79 | 35 |
| AIGS-GaS$_x$ | 518 | 36 | 68 | 36 |
|  | 568 | 40 | 71 | 37 |
|  | 528 | 31 | 53 | 39 |
|  | 543 | 37 | 99 | 39 |

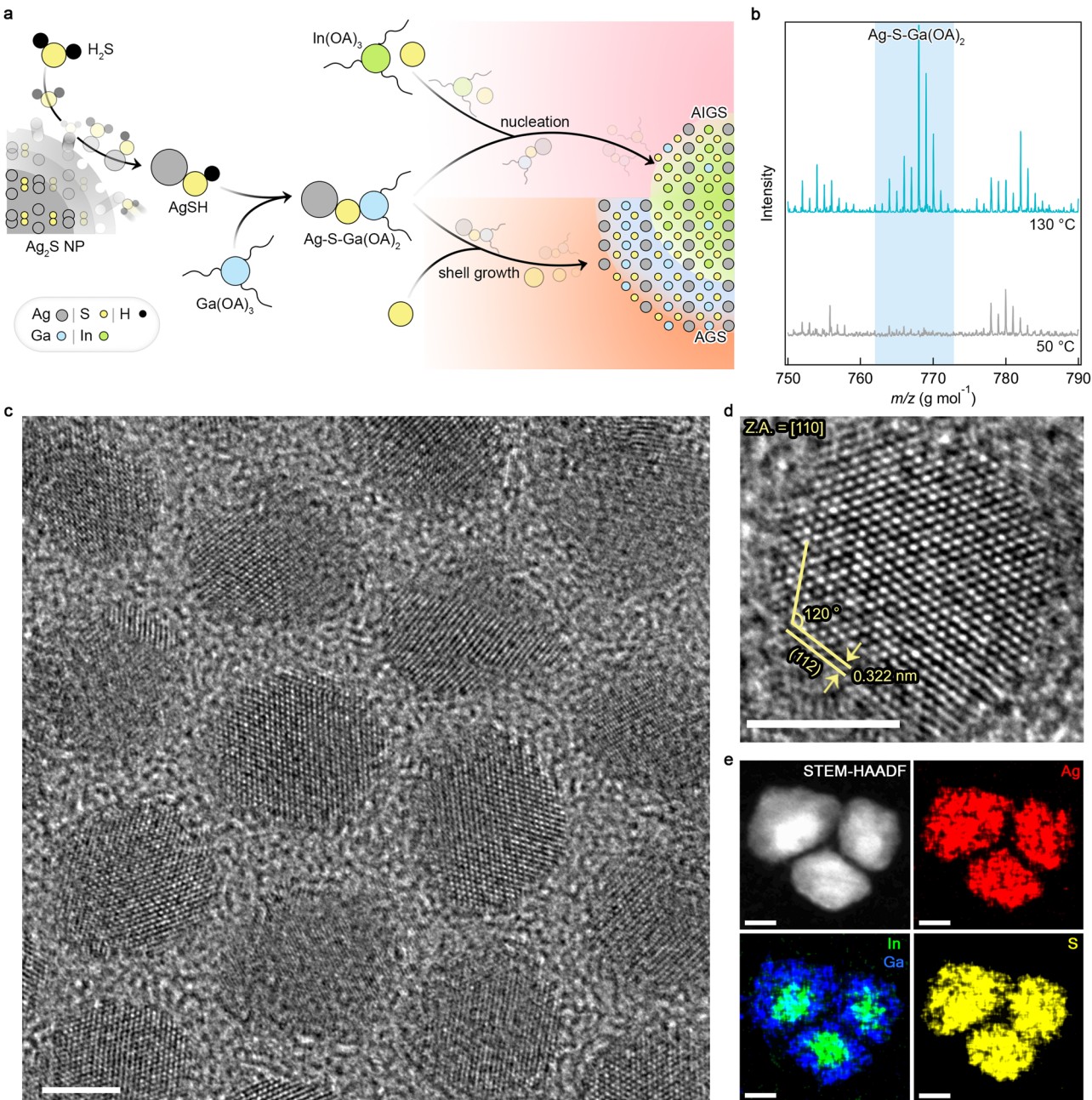

**Fig. 1 | AIGS-AGS core–shell NCs. a** Schematic illustration of AIGS-AGS NC synthesis. Key reactants and chemical intermediates are depicted. **b** MALDI-TOF mass spectra of the reactants containing Ag, S, Ga, and ligands (oleic acid (OA) and oleylamine (OAm)) taken at different reaction temperatures (130 °C (top, cyan) and 50 °C (bottom, gray)). The colored background highlights the presence of Ag–S–Ga(OA)$_2$ complex at the elevated temperature. The spectra are vertically shifted for visual clarity. **c**, **d** HR-TEM images of AIGS ($X = 0.5$, radius ($r$) = 1.8 nm)-AGS (shell thickness ($l$) = 2.9 nm) NCs. A NC is measured along the [110] axis. Z.A. means zone axis. **e** STEM-HAADF image and EDS elemental mapping of Ag (red), In (green), Ga (blue) and S (yellow) for AIGS ($X = 0.9$, $r = 2.95$ nm)-AGS ($l = 2.7$ nm) NCs. Scale bars in (**c**–**e**) are 5 nm. EDS analysis showing the compositional homogeneity among multiple individual AIGS-AGS NCs is presented in Supplementary Fig. 8. Source data are provided as a Source data file.

Fig. 10), while keeping the crystal structure (chalcopyrite) remains unchanged throughout the shell growth (Fig. 2c). AGS shell growth effectively confines the charge carriers in AIGS core and promotes radiative recombination, as shown in the enhancement of the optical transition between the lowest quantized states for electron and hole (1S$_e$–1S$_h$) (Fig. 2d) and ensemble PL decay dynamics (Supplementary Fig. 12). Particularly, the dominant broadband emission due to the surface defects when no shell is present is greatly suppressed upon AGS shelling. We observe that PL QY reaches to near unity with AGS shell thickness of 1.6–2.9 nm (Fig. 2d, inset), indicating the heteroepitaxy of AGS shell on AIGS core free from the formation of optically

active defects. The shell thickness greater than 3.0 nm accompanies a gradual decrease in PL QYs, which is attributed to the creation of internal defects by the accumulated structural stress between AIGS core and the thick AGS shell.

The composition of AIGS cores can be tuned by varying the molar ratio of injected In versus Ga contents (Fig. 2e–g and Supplementary Fig. 13). To obtain In content ($X$) in AIGS cores, we carry out elemental analysis (ICP-AES) or estimate it from XRD. Here, $X$ is [In]/([In]+[Ga]). The value of $X$ appears greater than the feed ratio, implying that In precursor indeed leads the nucleation (Supplementary Fig. 14). The energy gaps ($E_g = 1S_e–1S_h$) inferred from absorption spectra deviate

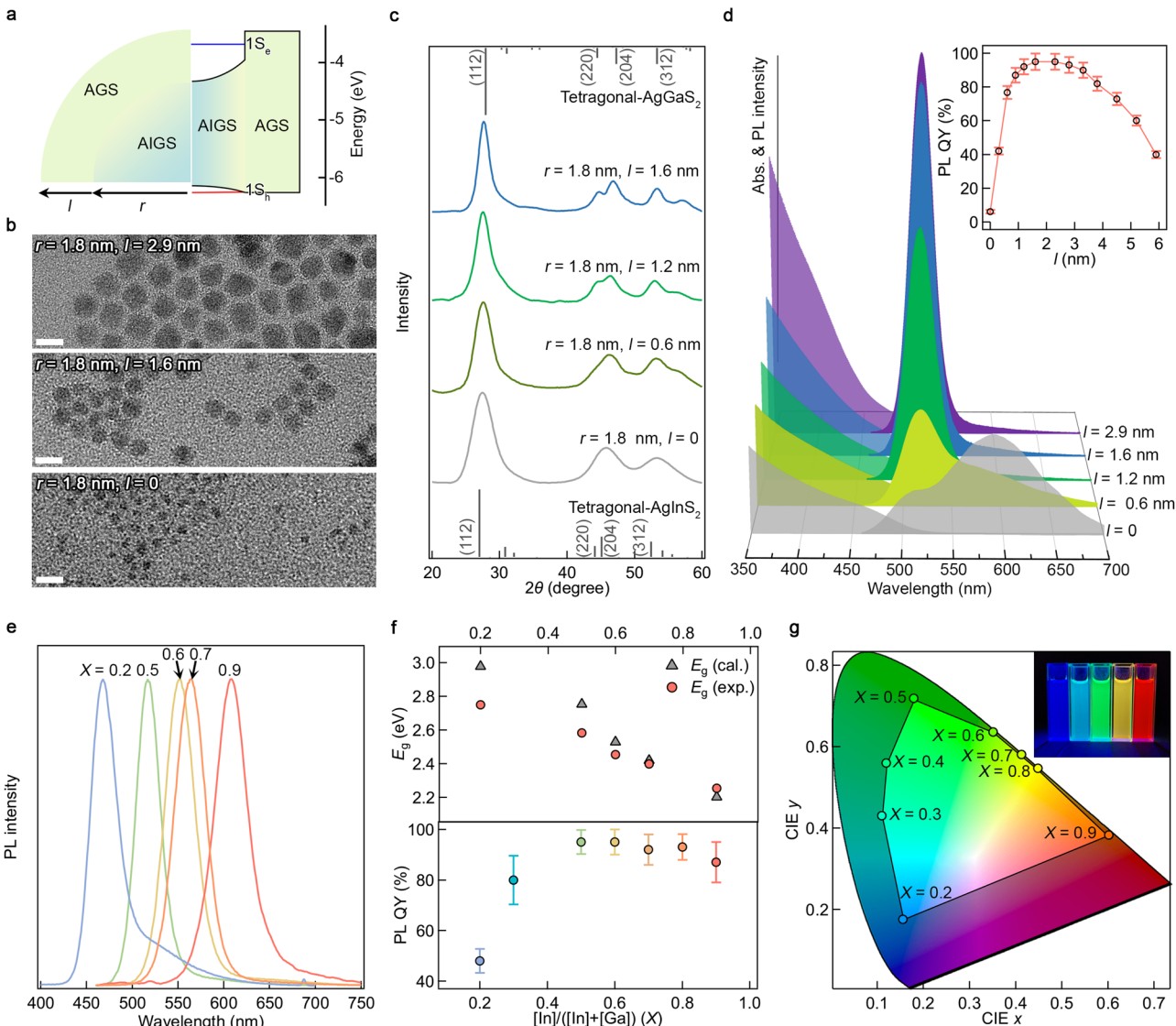

**Fig. 2 | AIGS-AGS NCs with variable core compositions and shell dimensions.**
**a** Schematic illustrations of the geometry (left) and potential profile (right) of AIGS-AGS NCs. Blue and red lines indicate the lowest quantized energy state for electron (1S$_e$) and hole (1S$_h$), respectively. **b** TEM images (scale bars = 10 nm), **c** X-ray diffraction (XRD) patterns, and **d** UV−Vis absorption and PL spectra of AIGS ($X = 0.5$, $r = 1.8$ nm)-AGS NCs with varying shell thicknesses ($0 \leq l \leq 2.9$ nm). 2d plots of PL spectra are supported in Supplementary Fig. 9. The inset displays PL QYs as a function of shell thickness. The average (symbols) and standard deviation (error bars) are from multiple independent runs of the experiment ($n = 30$). **e** PL spectra, **f** energy gaps (1S$_e$−1S$_h$) obtained from UV−Vis spectra (top) and PL QYs (bottom),

and **g** their color coordinates of AIGS-AGS ($l = 1.6$ nm) NCs with varying In contents ($0.2 \leq X \leq 0.9$) (inset: a photographic image of AIGS-AGS NCs with varying In ratios ($X = 0.2, 0.3, 0.5, 0.7,$ and $0.9$ from the left)). Gray triangles in (**f**) represent our calculation results for which homogenously alloyed AIGS cores are taken into account. The average (symbols) and standard deviation (error bars) in the bottom panel of (**f**) are from multiple independent runs of the experiment ($n = 10$). Synthesis and experimental results for AIGS-AGS NCs with varying compositions and shell dimensions are detailed in the Methods section, and Supplementary Figs. 7–16. Source data are provided as a Source data file.

from our quantum mechanical calculation where homogeneously alloyed AIGS cores and NCs' dimensions are taken into account, and the discrepancy becomes greater for NCs having lower In contents (upper panel of Fig. 2f, see "Methods" for a detailed explanation of calculation method). These coherently suggest that synthesized AIGS cores have composition gradients with In-rich interior and Ga-rich exterior. Nevertheless, resulting AIGS cores possess tetragonal crystal structure regardless of their compositions (chalcopyrite, Supplementary Fig. 13f), which promises the heteroepitaxy of tetragonal AGS shell onto them. The controllability of AIGS core compositions allows to expand the emission envelope of AIGS-AGS NCs to cover the entire visible region (Fig. 2e–g, see Table 2 for detailed optical properties of AIGS-AGS NCs). The capability for heteroepitaxy with AGS shell ensures high PL QYs. It is noted that AIGS-AGS NCs having higher Ga

contents ($X = 0.2$) show relatively low luminescence efficiencies (PL QY = 50%) due to the ineffective passivation of charge carriers by AGS shell (lower panel of Fig. 2f).

## Impact of AGS heteroepitaxy on photophysical properties

Two distinct PL emissions are present in AIGS cores, the narrowband emission (full width at half maximum (FWHM) = 30 nm) originating from 1S$_e$−1S$_h$ optical transition with a characteristic decay time ($\tau_{rad}$) of 40 ns and the broadband emission (FWHM of 40–70 nm) at lower energy positions with a slower recombination rate ($\tau_{rad} = 100$ ns) (Supplementary Fig. 17). The broadband emission is greatly suppressed by the AGS shell growth, implying that the broadband emission is related to the surface state of AIGS cores, rather than the compositional inhomogeneity among AIGS cores or the presence of

**Table 2 | Optical properties of AIGS-AGS NCs in this work**

| AIGS-AGS | | | | |
|---|---|---|---|---|
| X/r (nm)/l (nm)* | PL maximum (nm) | FWHM (nm) | PL QY (%) | Ref. |
| 0.5/1.8/0.6 | 518 | 55 | 77 | Fig. 2d |
| 0.5/1.8/1.2 | 517 | 34 | 91 | Fig. 2d |
| 0.5/1.8/1.6 | 517 | 31 | 96 | Fig. 2d |
| 0.5/1.8/2.9 | 517 | 30 | 96 | Fig. 2d |
| 0.2/1.8/1.6 | 468 | 34 | 49 | Fig. 2e |
| 0.6/2.2/1.6 | 551 | 38 | 95 | Fig. 2e |
| 0.7/2.3/1.6 | 565 | 39 | 92 | Fig. 2e |
| 0.9/2.9/1.6 | 610 | 42 | 86 | Fig. 2e |

*X/r/l represents In ratio in AIGS core [In]/([In]+[Ga])/AIGS core radius/AGS shell thickness.

chemical impurities in AIGS cores, which otherwise leave electronic footprints even after the passivation with AGS shell. The increase in PL QYs together with the suppression of the broadband emission in AIGS-AGS NCs signifies the effective passivation of the surface-related recombination channels.

The enhanced light emission of AIGS-AGS NCs can be also manifested in individual NCs, so we investigate the effect of AGS shell in single-dot spectroscopy. Figure 3a displays representative single-dot PL intensity traces for an AIGS core (lower panel) versus an AIGS-AGS ($l = 1.6$ nm) NC (upper panel). While the AIGS core suffers from severe PL blinking with on-time fraction of 11%, AIGS-AGS NC is highly emissive with on-time fraction of 77%. This implies that, in AIGS-AGS NCs, photoexcited charge carriers are effectively confined within AIGS core in Type I band structure and yield radiative recombination, instead of trapping in surface defects. Based on the analysis of individual NCs, the on-time fraction of individual NCs increases on average from 12% (no AGS shell) to 64% after 1.6 nm-thick AGS shell growth (Fig. 3b). A large NC-to-NC variation of the on-time statistics observed from AIGS-AGS NCs likely happens due to the change in the environment medium from solvent to air, which was unavoidable in the process of single-dot measurement.

In addition, for AIGS-AGS NCs, the surface-state related broadband emission is near-completely suppressed and the narrowband emission from $1S_e$–$1S_h$ transition is prevailing, as demonstrated in individual NC spectra in Fig. 3c, AIGS core (FWHM = 60 nm, upper panel) versus AIGS-AGS NC (FWHM = 17 nm, lower panel). The PL emission maximums of individual AIGS cores are distributed near the surface emission (≈600 nm) as observed in ensemble spectrum (gray trace in Fig. 2d) with a broad linewidth (≈48 nm on average). In a sharp contrast, the PL emission maximums of individual AIGS-AGS NCs are narrowly positioned near 520 nm, which is corresponding to the band edge emission, along with a very narrow linewidth (≈17 nm on average). All these findings are consistent with ensemble data and again corroborate the fact that the surface trap states of AIGS core are effectively passivated by AGS shell.

The effective passivation by AGS shell is also validated by the oxidative test (Fig. 3e). For the oxidate test, we monitor the change in PL QYs of NC solution upon exposure to air over time. The increase in the AGS shell thickness expands the half-lifetime of AIGS-AGS NCs, supporting that the width of potential wall (AGS shell thickness) is indeed responsible for the enhanced stability rather than the atomistic surface passivation. These results coherently attest that our approach, the heteroepitaxy of AGS shell on AIGS core, is an effective means to boost the photophysical and photochemical performance of NCs.

We note that the present approach, i.e., heteroepitaxy with I−III−VI₂ AGS shell, stands in sharp contrast to previous attempts to passivate the surface of AIGS cores using zincblende II−VI ZnS shells[34,35], amorphous GaS$_x$[36,37] or z-type ligands[38] regarding the

emission linewidth and efficiency (see Table 1 and Supplementary Fig. 1). Specifically, ZnS shell or z-type ligand aids to enhance PL QYs up to 70% by deactivating non-radiative recombination channels at the surface of AIGS cores but resulting NCs still entail strong broadband emission (≈100 nm). Amorphous GaS$_x$ shell eliminates the broadband emission of AIGS cores, but only marginal PL QY enhancement (≈70%) is allowed due to its inherent structural imperfection. Higher PL QY is achievable with the aid of delicate surface passivation with ligands[39]. By contrast, heteroepitaxy with AGS shell guarantees both bright and narrowband PL emission of AIGS-AGS NCs.

**Advantageous characteristics for photonic applications**

As demonstrated, AIGS-AGS NCs present a narrow emission linewidth and near-unity PL QY, promising their practical use in a range of photonic applications. To clarify the competitive advantages of AIGS-AGS NCs, we compare their optical properties with state-of-the-art heavy metal-free core−shell NCs (InP-ZnSeS NCs[25–28] and ZnSeTe−ZnSeS NCs[29–31]), both of which have similar PL energy (PL maximum ≈ 520 nm) and total dimension (diameter $d$ = 7.0 nm) (Fig. 4, see "Methods" and Supplementary Fig. 18 for detailed sample information). All NCs exhibit PL QYs over 90% and narrow size distribution ($\sigma < 15\%$), implying that the structural or compositional inhomogeneities are well controlled. However, AIGS-AGS NCs have a greater mass extinction coefficient than InP-ZnSeS NCs by a factor of 1.5−4 at the blue to UV-A region (Fig. 4a), leading to an important implication for down-conversion application under the excitation photon sources in blue to UV-A region. In addition, AIGS-AGS NCs show very narrow spectral linewidths in both ensemble and individual NC levels (Fig. 4b, c). Specifically, AIGS-AGS NCs have a narrower spectral linewidth (16.1 ± 1.6 nm) compared to InP-ZnSeS NCs (19.6 ± 1.6 nm) or ZnSeTe−ZnSeS NCs (27.8 ± 1.9 nm). We speculate that the narrow spectral linewidth of AIGS-AGS NCs is attributed to the effective confinement of charge carriers in Type I heterojunction potential profile. The narrow spectral linewidth of individual AIGS-AGS NCs is reflected to their collective PL spectra (FWHM of ensemble PL = 30 nm), which is much narrower than the reported spectral linewidths of ensemble InP-ZnSeS NCs (FWHM = 36 nm)[25,26,28] or ZnSeTe−ZnSeS NCs (FWHM = 41 nm)[29]. The narrower spectral linewidth and higher extinction coefficient of AIGS-AGS NCs promise a competitive edge over other heavy metal-free NCs in a range of photonic applications implementing down-converting NC layers (see Tables 1 and 2 and Supplementary Fig. 1). For example, the use of AIGS-AGS NCs in state-of-the-art NC displays in replace of InP-ZnSeS NCs awards not only offers an extended color space for images, but also a reduced thickness and weight, which are prerequisites for augmented reality or virtual reality displays.

The advantages in the form factors are of particular interest to LSCs that demand large-area down-converting NC films for an efficient harvest of sunlight. Figure 4d–f exemplify the competitiveness of AIGS-AGS NCs in LSCs. In LSCs, NC layer absorbs incident light and emits down-converted photons, which are waveguided to the edge, where the solar cell is mounted (Fig. 4d). The optical efficiency ($\eta_{opt}$) of LSC, defined by the ratio between incident photons through the top window and collected photons to the edge-mounted solar cells, is normally a function of incident photon absorption, re-emission efficiency of luminophore, reabsorption, and waveguiding efficiency of structure (see "Methods" for detailed theoretical model for LSCs)[40–42]. Thus, the desirable characteristics of luminophores are a high absorption coefficient at the wavelength of the incident light, high PL QY, and a large stoke shift. Our AIGS-AGS NCs fulfill the requirements of luminophores for LSCs at the point of absorption and re-emission efficiency (≈95%). Especially, the high absorption coefficient of AIGS-AGS NCs allows us to save the materials for LSCs by sufficiently absorbing incident light through dilute concentrated NCs. In addition, the small overlap between absorption and emission spectra of

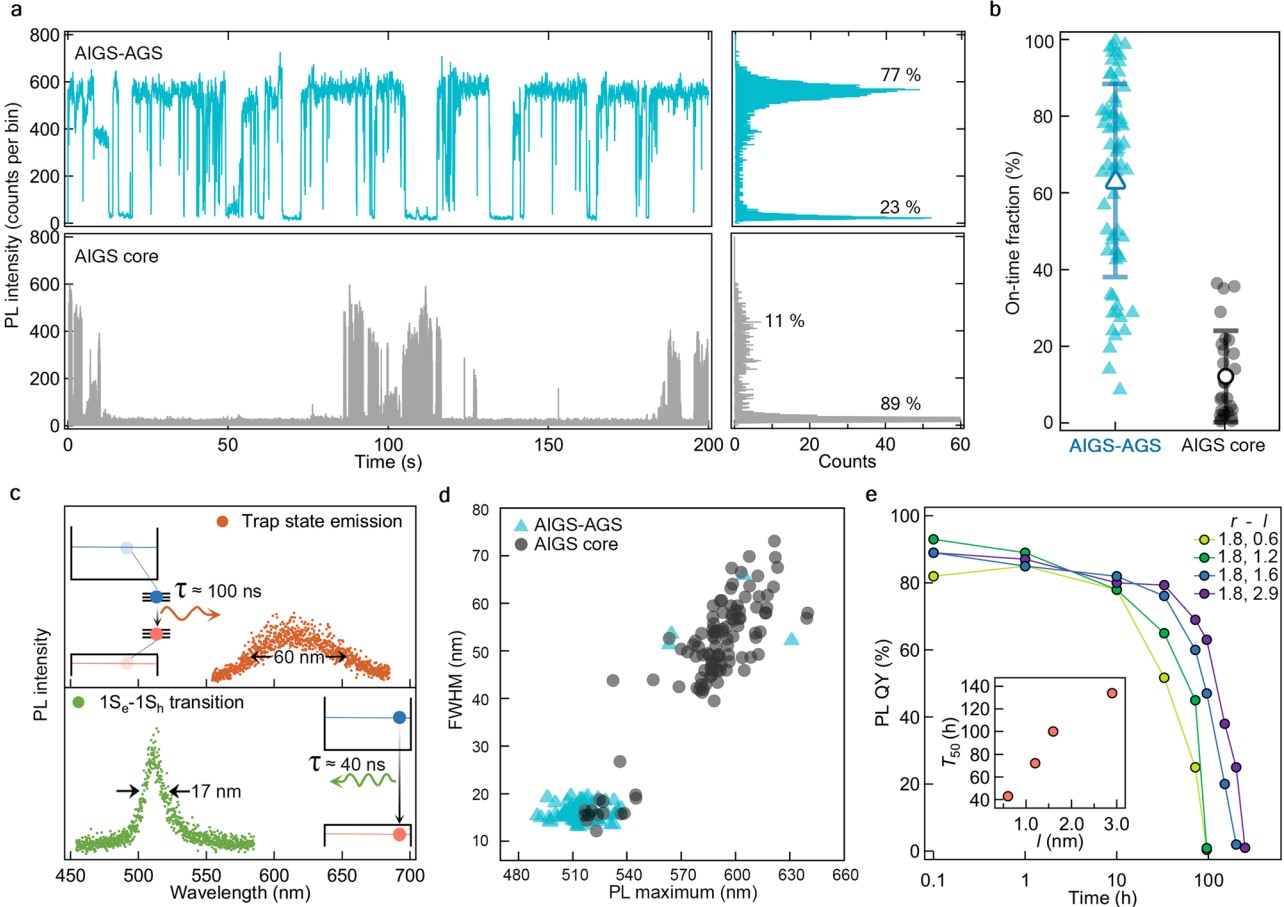

**Fig. 3 | Impact of AGS heteroepitaxy on photophysical and photochemical properties of individual AIGS-AGS NCs. a** Representative PL intensity trajectories (left) and PL intensity histograms (right) of AIGS-AGS NC (top, cyan) versus AIGS core (bottom, gray) (bin time = 50 ms). **b** On-time fraction statistics of individual AIGS-AGS NCs (left, cyan) and AIGS cores (right, gray). Average on-times are 63.9% and 12.3% for AIGS-AGS NCs and AIGS cores, respectively. The error bars in (**b**) represent the standard deviation ($n$ = 64 and 27 for AIGS-AGS and AIGS core, respectively). **c** Emission spectra from $1S_e$–$1S_h$ transition (bottom, AIGS-AGS NC) and surface trap states (top, AIGS core). The composition and dimensions are $X$ = 0.5, $r$ = 1.8 nm, and $l$ = 0 or 1.6 nm for AIGS cores or AIGS-AGS NCs, respectively. The insets depict luminescence mechanism and its characteristic time. **d** PL maximum versus FWHM distribution in individual AIGS-AGS NCs or AIGS cores. **e** Time-dependent PL QYs of AIGS ($X$ = 0.5, $r$ = 1.8 nm)-AGS NCs with varying shell thicknesses ($l$ = 0.6, 1.2, 1.6, or 2.9 nm) upon oxidative test. The inset displays $T_{50}$, the time when PL QY reaches to the half of initial values. See Supplementary Fig. 17 for detailed optical characteristics (exciton lifetimes and $g^{(2)}(t)$) of individual NCs. Source data are provided as a Source data file.

AIGS-AGS NCs (shown in Fig. 2d and Supplementary Fig. 9) suppresses the reabsorption loss in the LSC, thereby leading to the improved efficiency.

Regarding these factors, we compared the performance of LSC with AIGS-AGS NC solution to one employing InP-ZnSeS NC solution of various concentrations. Here, $\eta_{opt}$ of LSC is derived from the short circuit current of edge-mounted crystalline silicon solar cell ($10 \times 50$ mm$^2$) and incident light intensity (shown in Fig. 4e and Supplementary Fig. 19). As more luminophores are included in LSC, its $\eta_{opt}$ increases typically at a lower concentration of luminophore due to enhanced light absorption. Thus, $\eta_{opt}$s of dilute concentrated LSCs with InP-ZnSeS and AIGS-AGS NCs are correlated with their concentration. Despite similar trends in the concentration and $\eta_{opt}$, the LSC employing AIGS-AGS NCs exhibits higher $\eta_{opt}$ than LSC with InP-ZnSeS NCs at a low concentration (Fig. 4f). High absorption coefficient of AIGS-AGS NCs provides better absorption of incident photons, resulting in the increased $\eta_{opt}$. Consequently, the luminophore concentration for maximized $\eta_{opt}$ is much lower in the case of AIGS-AGS NCs compared to the case of InP-ZnSeS NCs. According to the theoretical simulation and experiments, the optimized concentration of LSC with AIGS-AGS NCs is around 0.5 mg mL$^{-1}$ (see ref. 42), whereas 1 mg mL$^{-1}$ of InP-ZnSeS NCs is required to reach the maximum $\eta_{opt}$.

Consistently with the expectation in LSC performance from the optical properties of AIGS-AGS NCs, the LSC experiment implies that AIGS-AGS NCs are strong candidates to demonstrate thin LSCs.

Thin LSCs would be beneficial to reduce fabrication costs and minimize material waste for solar energy harvesting and optoelectronic devices. Besides, the expected advantage from LSCs implementing AIGS-AGS NCs is enhanced mechanical strength, which plays a critical role in flexible and wearable devices. Given that the tensile and strain applied to the thin film is much smaller than those to the thick film[43], flexible LSC with a thin absorption layer will be more robust to the deformation of its shape, which is suitable to cover curved surfaces. Moreover, it would be helpful for the rigid LSC, expected to cover building envelope. Field-installed LSC will be frequently exposed to harsh outdoor conditions, where the strain and tensile will be applied continuously due to temperature changes throughout the year. If the LSC consists of a thin absorption layer, delamination and crack from thermal-mechanical stress will be reduced. In these respects, AIGS-AGS NCs not only ensure higher performance of LSCs but also contribute to realizing economic, environmental, and reliable LSCs.

We conclude by noting that the photophysical properties of AIGS-AGS NCs are in stark contrast to those of its twin compound CuInS$_2$

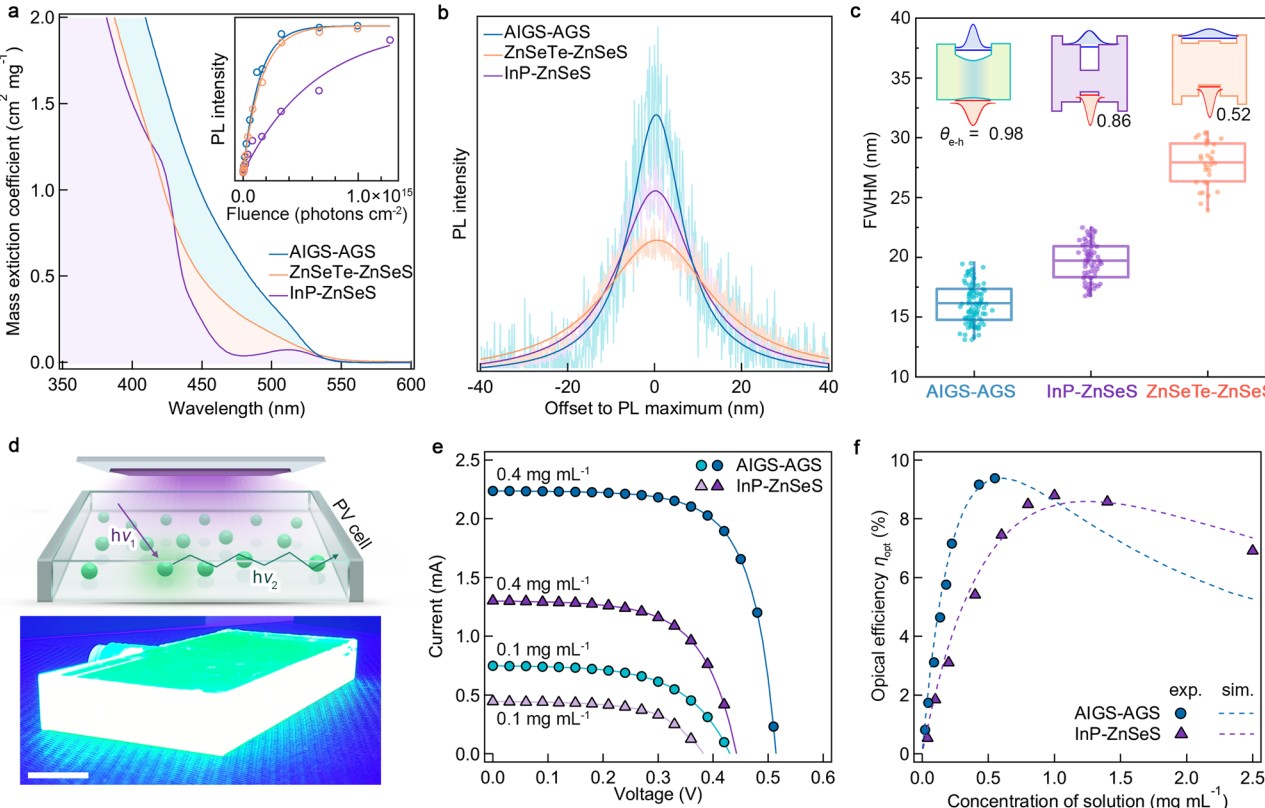

**Fig. 4 | Competitive advantages of AIGS-AGS NCs and their application to LSC.**
**a** Mass extinction coefficients, **b** representative PL spectra in individual NC level, and **c** FWHM statistics of AIGS-AGS, ZnSeTe–ZnSeS, and InP-ZnSeS NCs. The inset in (**a**) shows pump-fluence-dependence of the single exciton PL intensities to gain optical absorption cross-sections ($\sigma$) of NCs @ 450 nm ($\sigma$ = 8.15, 6.54, and 1.58 ($10^{-15}$ cm$^2$) for AIGS-AGS, ZnSeTe–ZnSeS, and InP-ZnSeS NCs, respectively). Single exciton PL amplitudes from pump-fluence dependence measurements are fitted based on Poisson statistics of photon absorption to gain optical cross-sections. The PL spectra in (**b**) are fitted with single Lorentzian function and normalized by their integrated areas. The box-and-whisker plots in (**c**) represent the distribution of the data. ($n$ = 103, 71, and 31 for AIGS-AGS, InP-ZnSeS, and ZnSeTe–ZnSeS, respectively). The insets in (**c**) display the band alignment of each heterostructured NC

and the electron-hole overlap integral ($\theta_{e-h}$). **d** Schematic and photographic image of a LSC, whose size is $100 \times 50 \times 10$ mm$^3$. The incident photons from top of LSC ($50 \times 100$ mm$^2$) are absorbed, re-emitted, and waveguided to the edge-mounted solar cell ($50 \times 10$ mm$^2$). Here, the geometric factor of LSC is 10, and light source is 420 nm LEDs. Scale bar in (**d**) is 10 mm. **e** Current-voltage characteristics of edge-mounted c-Si cells with 0.1 and 0.4 mg mL$^{-1}$ of AIGS-AGS (circle) or InP-ZnSeS (triangle) NC solutions. **f** $\eta_{opt}$s of LSCs with varying concentrations of AIGS-AGS and InP-ZnSeS NC solution (exp.) with a theoretical simulation (sim.)[42]. Improved light absorption of AIGS-AGS NCs allows us to obtain higher efficiency of LSC with reduced NC concentrations or film thicknesses. Synthesis and characteristics of ZnSeTe–ZnSeS and InP-ZnSeS NCs are detailed in "Methods" and Supplementary Fig. 18. Source data are provided as a Source data file.

(CIS) NCs, thereby expanding the application scope of I–III–VI$_2$ NCs. CIS NCs exhibit high PL QYs in the green to near-IR region (PL maximum energy from 1.5 to 2.4 eV) with large Stokes shift (300–500 meV), broad emission linewidth (FWHM > 400 meV), and delayed radiative decay time ($\approx$200 ns) due to Cu hole trap states[44,45] from two different oxidation states of Cu, namely cuprous (Cu$^+$) and cupric (Cu$^{2+}$). These characteristics guarantee a wide range of solar spectrum absorption and the suppression of energy loss via reabsorption processes, which are best adapted for high-efficiency LSCs[20,46–49]. AIGS-AGS NCs, in contrast, exhibit high PL QYs in the visible region (PL maximum energy from 2.0 to 2.6 eV) with narrow emission linewidth (FWHM $\approx$ 140 meV) and rapid radiative decay time ($\approx$ 40 ns), making them suitable for use in displays, lightings, and colorful LSCs[24,36–39]. In particular, the LSC with AIGS-AGS NCS will provide esthetic appealing to energy harvesting system incorporating architectures with small losses[22,50–52].

In summary, we have presented environmentally benign nano-emitters made of AIGS-AGS I–III–VI$_2$–I–III–VI$_2$ core–shell heterostructures. We have designed a chemical scheme that deploys Ag–S–Ga(OA)$_2$ complex for both AIGS core synthesis and AGS shell growth. The heteroepitaxy of AIGS and AGS constructs Type I potential profile that effectively confines charge carriers within the emissive core without the creation of optically active defects, allowing for near-

unity PL QY and photochemical stability. Resulting AIGS-AGS NCs show higher extinction coefficient and narrower spectral linewidth compared to state-of-the-art heavy metal-free NCs, promising their immediate use in a range of practicable photonic applications.

## Methods

### Materials

Silver (I) iodide (AgI, 99.9%), tellurium (Te, 99.999%) was purchased from Alfa aesar. Indium (III) acetate (In(ac)$_3$, 99.99%), sulfur powder (S$_8$, 99.9%), oleic acid (OA, 99%), 1-octadecene (ODE, 99%), tris(trimethylsilyl)phosphine ((TMS)$_3$P, 99.9%), selenium (Se, 99.9%), and $n$-trioctylphosphine (TOP, 99%) were purchased from Uniam. Gallium (III) acetylacetonate (Ga(acac)$_3$, 99.99%), oleylamine (OAm, 70%), 1-dodecanethiol (DDT, ≥98%), diphenylphosphine (DPP, 98%), ethanol (>99.8%), and all solvents including toluene (anhydrous 99.8%), ethanol (anhydrous ≥99.5%) were purchased from Sigma-Aldrich. Gallium chloride (GaCl$_3$, 98%) was purchased from TCI. All chemicals, unless otherwise stated, were used as received.

### Precursor preparation

All chemistry is conducted with Schlenk line technique. 0.5 M indium oleate (In(OA)$_3$), gallium oleate (Ga(OA)$_3$) and zinc oleate (Zn(OA)$_2$) stock solutions in ODE are prepared for cation precursors and 0.5 M

sulfur dissolved in OAm (S-OAm), 2 M TOPS, 2 M TOPSe, 0.5 M TOPTe and 0.2 M DPPSe are prepared for anion precursors. For In(OA)$_3$ preparation, 50 mmol of In(ac)$_3$ and 150 mmol of OA are mixed in a three-neck flask (250 mL), degassed at 130 °C for 6 h, backfilled with N$_2$, and diluted to 0.5 M concentration with ODE. Ga(OA)$_3$ and Zn(OA)$_2$ are prepared in the same procedure with Ga(acac)$_3$ and Zn(ac)$_2$ in replace of In(ac)$_3$, respectively. For S-OAm stock solution, 50 mmol of sulfur powder is mixed in 100 mL of OAm, degassed at room temperature for 1 h and backfilled with N$_2$ for further reaction. For TOPS preparation, 100 mmol of sulfur powder is mixed with 50 mL of TOP at 100 °C for 6 h in the glovebox. The same procedure is followed to prepare TOPSe and TOPTe. For DPPSe, 4 mmol of Se is mixed with 2 mL of DPP at 200 °C for 5 min in the glovebox, and diluted to 0.2 M concentration with toluene at RT.

## Ag–S-Ga(OA)$_2$ preparation

All synthesis was carried out under N$_2$ atmosphere through the Schlenk line technique. For preparing Ag$_2$S NPs, 1 mmol of AgI and 5 mL of OAm were loaded in a three-neck flask and degassed at 50 °C for 1 h. After the flask was backfilled with N$_2$, 2.5 mL of DDT and 1 mL of 0.5 M S-OAm were injected into the reaction flask to form Ag$_2$S NPs. To transform Ag$_2$S NPs to Ag–S–Ga(OA)$_2$, 4.5 mL of 0.5 M Ga(OA)$_3$ and 8 mL of 0.5 M S-OAm were added into the reaction flask and the temperature was elevated to 210 °C. We note that excess Ga(OA)$_3$ and S were deployed for the complete conversion of Ag$_2$S NPs to Ag–S–Ga(OA)$_2$, and unreacted Ga(OA)$_3$ and S were used for AIGS core synthesis.

## AIGS core synthesis

Reaction flask containing 1.5 mL of Ag–S–Ga(OA)$_2$ stock solution and 5 mL of OAm is filled with N$_2$ and heated up to 210 °C. In all, 0.15 mL of 0.5 M In(OA)$_3$ precursor were swiftly injected into the flask for the nucleation and subsequent growth of AIGS cores ($X = 0.5$, $r = 1.8$ nm). We note that the addition of extra S precursor is not needed because excess S-OAm in Ag–S–Ga(OA)$_2$ stock solution spontaneously participates in the reaction. The reaction temperature was maintained at the elevated temperature for 30 min and cooled to RT to cease the reaction. The injected volume of In(OA)$_3$ precursor was controlled to vary the In content ($X$) in AIGS cores. Specifically, 0.05, 0.3, 0.6, and 2.7 mL of 0.5 M In(OA)$_3$ precursor yielded AIGS NCs with In contents ($X$) of 0.2, 0.6, 0.7, and 0.9, respectively. The resulting AIGS cores were purified twice in the glovebox by precipitation (ethanol)/redispersion (toluene) method at 6000×$g$ for 5 min and finally dispersed in 5 mL of toluene.

## AGS shell growth

A reaction flask containing 5 mL of OAm and 300 mg of AIGS cores ($X = 0.5$) was degassed at 110 °C for 1 h, backfilled with N$_2$ and heated up to 240 °C for AGS shell growth. At the elevated temperature, 0.3 mL of Ag–S–Ga(OA)$_2$ precursor solution and 0.15 mL of 0.5 M S-OAm were injected, and the reaction temperature was maintained for 1 h to grow 0.3-nm-thick AGS shell. Alternatively, the equivalent amount of Ag$_2$S NPs, Ga(OA)$_3$ and S-OAm solution were added for AGS shell growth. It is noted that when shell thickness exceeded 0.6 nm, 10 M GaCl$_3$ solution (dissolved in ethanol, 1 eq of Ga(OA)$_3$) was added to facilitate uniform shell growth[39]. For thicker AGS shell growth, Ag, Ga, and S precursors were injected repeatedly to grow AGS shell (0.3–0.5 nm for each step) at the fixed reaction temperature (240 °C). An excess amount of S compared to Ag or Ga is deployed throughout the reaction for higher PL QY and suppressed trap emission of AIGS-AGS NCs. The resulting AIGS-AGS NCs were purified twice in glovebox by precipitation (ethanol)/redispersion (toluene) method at 6000×$g$ for 5 min and finally dispersed in 5 mL of toluene for further characterization and applications. Changes in the optical properties of NCs during the repeated purification steps are provided in Supplementary Fig. 11.

## ZnSeTe–ZnSeS NC (ZnSe$_{0.67}$Te$_{0.33}$ ($r = 1.8$ nm)-ZnSe ($l = 1.2$ nm)-ZnS ($h = 0.6$ nm)) synthesis

ZnSeTe–ZnSeS NCs are synthesized following the previously reported method[29] with minor modifications. In all, 1.2 mL of 0.5 M Zn(OA)$_2$ and 15 mL of ODE were stirred and degassed at 110 °C in a 3-neck round flask. After 1 h of degassing to remove water and oxygen completely, it was filled with N$_2$. Then, 1 mL of 0.2 M DPPSe and 0.2 mL of 0.5 M TOPTe was injected at 230 °C were injected to synthesize ZnSe$_{0.67}$Te$_{0.33}$ cores at 230 °C and maintained for 30 min. After that, heated to 300 °C for 15 min to grow ZnSe$_{0.67}$Te$_{0.33}$ cores completely ($r = 1.8$ nm). For further shell growth on the seed/emissive layer, 2 mL of 0.5 M Zn(OA)$_2$ and 0.25 mL of 2 M TOPSe and 3.4 mL of 0.5 M Zn(OA)$_2$ and 0.425 mL of TOPSe were injected in order and maintained at 320 °C for 1 h. Additional injection of 5.0 mL of 0.5 M Zn(OA)$_2$ and 0.2 mL of DDT results in 0.6 nm-thick ZnS shell growth. Absorption, PL spectra, characterization, and TEM image of ZnSe$_{0.67}$Te$_{0.33}$-ZnSeS NCs are provided in Supplementary Fig. 18.

## InP-ZnSeS NC (InP ($r = 1.2$ nm)-ZnSe ($l = 1.8$ nm)-ZnS ($h = 0.5$ nm)) synthesis

InP-ZnSeS NCs are synthesized following the previously reported method[26] with minor modifications. For InP cores, 60 mL of 0.5 M of In(OA)$_3$ and 400 mL of ODE were degassed at 110 °C for 1 h, filled with Ar. 15 mmol of P(TMS)$_3$ diluted with 30 mL of TOP was injected into the reaction flask to form indium-phosphine complex. The reaction flask was then heated up to 260 °C in 15 min to activate InP nucleation. The reaction temperature was maintained for 1 h and decreased to room temperature to complete the reaction. Synthesized InP cores were purified twice by precipitation (acetone)/redispersion (toluene) method at 6000× $g$ for 5 min and finally diluted at a concentration of 200 mg mL$^{-1}$ (in toluene) and kept in refrigerator for further experiments. For InP-ZnSeS NCs, 10 mL of 0.5 M of Zn(OA)$_2$ and 10 mL of TOA are degassed at 110 °C for 1 h, filled with Ar and heated up to 180 °C. 100 mg of InP ($r = 1.2$ nm) NCs is injected to the reaction flask, and 1.4 mL of 2 M TOPSe precursor is added to promote ZnSe shell growth. The temperature is elevated to 320 °C and kept for 2 h. 20 mL of Zn(OA)$_2$ precursor, and 3.1 mmol of TOPS are added stepwise for 2 h to grow 0.5 nm ZnS shell. Absorption, PL spectra, characterization, and TEM image of InP-ZnSeS NCs are provided in Supplementary Fig. 18.

## Characterization

UV–Vis, PL and PL QY measurements were conducted with UV-1800 (Shimadzu), FluoroMax-4 (Horiba), and quantaurus-QY plus (Hamamatsu Photonics), respectively. HR-TEM images were obtained with Talos F200i working at 200 kV. The crystalline structures of NCs were investigated with XRD at the 5 A beamline ($\lambda = 0.154$ nm) of the Pohang Accelerator Laboratory (PAL). Time-resolved emission spectra measurement was conducted using the 405 nm (3.06 eV) excitation beam at 500 kHz repetition rate (PicoQuant, LDH-D-C-405 laser diode) with photo multiplier tube (PicoQuant, PMA-C182-N-M) coupled to the monochromator (Teledyne Princeton Instruments, SpectraPro SP-2150). Single-dot measurements were conducted using Hanbury Brown-Twiss setup with the 450 nm (2.76 eV) excitation beam (PicoQuant, LDH-D-C-450 laser diode). The laser beam was focused on the sample using the oil immersion objective (Olympus, UPLXAPO100XO, NA 1.45) and signals were collected through the same objective lens and directed to single-photon avalanche diodes (Micro Photon Devices, PDM Series) connected with time-correlated single-photon counting module (PicoQuant, HydraHarp 400) or to EMCCD camera (Princeton Instruments, ProEM HS1024BX3) attached to the spectrometer (Princeton Instruments, IsoPlane SCT320). The chemical composition of AIGS was analyzed by ICP-AES (Perkin-Elmer, OPTIMA-4300DV). The temperature-dependent PL characteristics of solution NC samples were acquired with EMCCD camera (Princeton

Instruments, ProEM HS1024BX3) under excitation with 450 nm lamp (CoolLED, pE-300white series SB) (Supplementary Fig. 20). MALDI-TOF mass spectroscopy was conducted with Voyager DE-STR (Applied Biosystems). The analyte and matrix (dithranol) were mixed in chloroform. The dried samples were ionized by pulsed $N_2$ laser (337 nm, 3 ns pulses). Overall, 20.0 kV was applied for accelerating positive ions. The reflector mode was used for MS analysis.

### Energy gap calculation
The exciton energy of the charged NC is calculated by custom written python code where two band K dot P model is implemented[53,54]. The finite difference scheme converts $\mathbf{K} \cdot \mathbf{P}$ matrix calculation to linear algebra with equally spaced 1.5 Å mesh. We applied cylindrical symmetry for the Laplacian operator so that full 3d simulation is achieved. To be a self-consistency, Schrödinger and Poisson equations are solved iteratively until the energy difference ($\Delta E$) is less than 1 meV. The compositions and radius of AIGS cores experimentally obtained from ICP-AES and TEM (Supplementary Fig. 14) are considered for the exciton energy calculation.

### Dissociation energy calculation
Density functional theory (DFT) calculations were performed to obtain the bond-dissociation energies. The projector augmented wave (PAW) potentials[55] were used as implemented in Vienna ab-initio Simulation Package (VASP)[56]. The exchange-correlation function suggested by using Perdew, Burke, and Ernzerhof was used[57]. The energy cutoff for the plane waves was set to 400 eV, and the atoms were relaxed until the residual force became below 0.01 eV Å$^{-1}$. The bond strength between a metal atom oleates (OA) was estimated by employing a metal atom (Al, Ga, or In) bonded with three acetate (CH3COO) molecules. The bond-dissociation energy of a metal and an oxygen atom was calculated by detaching one of the acetates from the metal. As summarized in Supplementary Table 1, an In atom required the lowest energy to break the metal-oxygen bond, and the bond becomes stronger as the atom changes from In to Ga and Al.

### LSC fabrication and characterization
InP-ZnSeS NCs and AIGS-AGS NCs with similar PL QY (≈95%) were used for LSCs. LSCs with NC dispersion in toluene were loaded in a clear quartz cuvette ($100 \times 500 \times 10$ mm$^3$) to avoid inevitable PL QY drop during the polymerization process. The efficiency of LSC was monitored by a crystalline silicon solar cell ($10 \times 50$ mm$^2$) mounted at one edge of LSC without index-matching oil, while the metal-based reflective films were installed on the other three sides. The top and bottom sides of LSC were enclosed by clear quartz, whose light trapping efficiency is 0.75 due to the refractive index difference between air and the quartz glass. The efficiency and absorptance of LSC with different concentrated NCs were repeatedly evaluated under 420 nm LED sources (1.4 mW cm$^{-2}$, $3.0 \times 10^{15}$ photons cm$^{-2}$).

### Theoretical model for LSC based on optical properties of NCs
Theoretical optical efficiencies of LSC with AIGS-AGS and InP-ZnSeS NCs were calculated based on previously suggested models[40,42]. The optical efficiency ($\eta_{opt}$) of LSC can be expressed following:

$$\eta_{opt} = \frac{(1-R)(1-e^{-<\alpha_1>})\eta_{PL}\eta_{tr}}{1+4\beta G<Q_{LSC}>^{-1}(1-\eta_{PL}\eta_{tr})<\alpha_1>} \tag{1}$$

Where $R$ is the reflectance of LSC to the incident light, $<\alpha_1>$ is the average absorbance of NCs at incident light, $\eta_{PL}$ is PL QY of NCs (≈0.95), $\eta_{tr}$ is waveguiding efficiency at the interface between quartz glass and air (≈0.75), $\beta$ is constant, $G$ is a geometric factor (≈10) and $<Q_{LSC}>$ is the average quality factor. Here, the $<Q_{LSC}>$ is the ratio of $<\alpha_1>$ and $<\alpha_2>$, where $<\alpha_2>$ is the average absorbance of NCs under the emission

spectrum of NCs, as follows.

$$<Q_{LSC}> = \frac{<\alpha_1>}{<\alpha_2>} \tag{2}$$

$$<\alpha_1> = \frac{\int \alpha_1(\lambda)\Phi_{LED}(\lambda)}{\int \Phi_{LED}(\lambda)} \tag{3}$$

$$<\alpha_2> = \frac{\int \alpha_2(\lambda)\Phi_{PL}(\lambda)}{\int \Phi_{PL}(\lambda)} \tag{4}$$

where, $\Phi_{LED}(\lambda)$ and $\Phi_{PL}(\lambda)$ are flex density spectrum of excited LED and PL of luminophore. The calculated $<Q_{LSC}>$ of AIGS-AGS and InP-ZnSeS NCs are 18.48 and 15.27, respectively. Due to the developed 1 s peak of InP-ZnSeS NCs, its $<\alpha_2>$ is higher than that of AIGS-AGS NCs. We believe that improved $<Q_{LSC}>$ of AIGS-AGS NCs leads to better $\eta_{opt}$ in the experiment and theoretical analysis.

### Reporting summary
Further information on research design is available in the Nature Portfolio Reporting Summary linked to this article.

## Data availability
The data that support the findings of this study are available from the corresponding authors upon request. Source data are provided with this paper.

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

## Acknowledgements

This work was supported by the National Research Foundation of Korea (NRF) funded by the Ministry of Science, ICT and Future Planning (2020R1A2C2011478, 2020M3D1A2101310, 2021M3H4A3A01062960, 2021M3H4A1A01004332, and 2022R1A2C1092582), the Ministry of Trade, Industry & Energy (MOTIE, Korea) (20010737 and 20019417) and Samsung Display. H.J.L. and K.C. acknowledge SAIT, Samsung Electronics Co., Ltd.

## Author contributions

W.K.B., J.H.C., H.J.L., and K.C. conceived the idea. H.J.L., S.I., K.K., J.A.C., J.L., J.W.P., D.S., B.G.J., and J.H.C. conducted synthesis and structural characterization. D.J.J., Y.S.P., and D.C.L. led the spectroscopic analysis. J.S.P., E.H., and J.H.C. performed computational calculations. H.J.S. carried out LSC fabrication and characterization. All authors contributed to the manuscript preparation.

## Competing interests

The authors declare no competing interests.

## Additional information

[1]SKKU Advanced Institute of Nanotechnology (SAINT), Sungkyunkwan University (SKKU), Suwon 16419, Republic of Korea. [2]School of Chemical and Biological Engineering, Seoul National University, Seoul 08826, Republic of Korea. [3]School of Chemical and Biomolecular Engineering, Pusan National University, Busan 46241, Republic of Korea. [4]Department of Chemical and Biomolecular Engineering, KAIST Institute for the Nanocentury, Korea Advanced Institute of Science and Technology (KAIST), Daejeon 34141, Republic of Korea. [5]Chemistry Division, Los Alamos National Laboratory, Los Alamos, NM 87545, USA. [6]Department of Safety Engineering, Seoul National University of Science and Technology, Seoul 01811, Republic of Korea. [7]Department of Chemistry, James Franck Institute, and Pritzker School of Molecular Engineering, University of Chicago, Chicago, IL 60637, USA. [8]These authors contributed equally: Hak June Lee, Seongbin Im, Dongju Jung. ✉e-mail: hj.song@seoultech.ac.kr; junhyuk@uchicago.edu; wkbae@skku.edu

