## [Peer review file · Nature Communications]

REVIEWER COMMENTS

Reviewer #1 (Remarks to the Author):

This work by Lee et al. reports on the synthesis of core shell Ag(In,Ga)S₂/AgGaS₂ nanocrystals for optoelectronic applications. The interest in I-III-VI₂ semiconductors is timely in particular to replace heavy metal based compounds as light emitters. The chemistry of I-III-VI₂ nanocrystals and in particular the fine control of their surface defects is, however, substantially more complicated than common II-VI or III-V systems resulting in typically broad emission lines and lower emission yields. The authors tackle this challenge through an heteroepitaxy approach that results in type I band alignment, near unity emission yield and narrow emission spectra.

The chemical/photophysical results are interesting but the application perspectives, in particular for luminescent solar concentration (LSCs), do not seem fully justified in particular with respect to the state of the art. The way some data is reported (e.g. Fig2d) makes it hard to extract the relevant information (absorption/emission overlap integral). Overall the paper could be suitable for Nature Comm but some amendments are necessary.

More specifically:

- The achievement of near unity photoluminescence yield and narrow emission with I-III-VI₂ system is highly relevant also for better understanding the physics of twin compounds such as CuInS₂ or AgInS₂ nanocrystals for which the single particle emission spectrum is still largely debated (see Gamelin vs Klimov results on single CuInS₂ QD spectroscopy). The authors focus on the chemistry and do not discuss their findings in this framework which is actually quite relevant. I strongly suggest to do so and possibly discuss the possible emission mechanism of their systems which is now completely lacking. This would make the paper much more interesting and complete and offer the community a new valuable piece of information. There are also important works on the emission mechanisms in AgInS₂ and CuInS₂ that are not cited and would help to contextualize the results better.

- It is impossible to evaluate the spectral overlap between the absorption and PL spectra of shelled crystals from Fig 2d, but based on the position of the excitonic peak for the bare nanocrystals, the spectral overlap seems to be comparable with the exciton emission in III-V nanocrystals. This is interesting from a fundamental perspective but is a technological problem as III-Vs are unsuitable for solar concentration because of excessive self-absorption. The authors should re-plot figure 2 and evaluate the LSC applicability based on quantitative figures (eg. Stokes shift, effective overlap for a given nanocrystal concentration ecc.). There are relevant reports on LSCs that are not cited that would help this analysis.

- The comparison with ZnTe and InP nanocrystals for LSCs is irrelevant as neither are suitable LSC emitters. The comparison should be made with AgInS₂ or CuInS₂ systems that represent the state of the art in the field and needs to be done much more rigorously, for example by comparing light propagation for a given solar absorptance.

- One suggestion for the authors that might add to the appeal of their results is to test the effect of temperature on the emission properties of their nanocrystals. Phonon coupling is a critical aspect in the application of III-V nanocrystals as light converters (both white light and color displays) as it quenches the luminescence and changes the emission color. The core/shell particles reported here could benefit from heterostructuring in both aspects and result substantially better than the state of the art in the field.

Reviewer #2 (Remarks to the Author):

Reviewer's remarks on the manuscript "Coherent Heteroepitaxial Growth of I-III-VI₂ Ag(In,Ga)₂S₂ Colloidal Nanocrystals with High Absorption Cross-Section and Near-unity Quantum Yield"

The manuscript reports on highly emissive core/shell quantum dots with Ag-In-Ga-S (AIGS) cores and Ag-Ga-S (AGS) shells produced by using a general and new precursor. A molecular precursor combining Ag, Ga, and S is proposed that allows to equalize the reactivity of Ag, In, and Ga and achieve very homogeneous and controlled Ag-In-Ga-S NC nucleation as well as subsequent formation of a protective Ag-Ga-S shell.

The AIGS/AGS QDs show unprecedentedly high PL QY reaching almost unity along with a very low spectral widths of PL bands, which have never been reported so far for ternary I-III-VI QDs. Single-QD studies showed a high suppression of blinking and non-emissive states in cores due to epitaxial deposition of the shell which has a minimal lattice mismatch with the core. Both ultimate PL QYs and very low FWHMs of PL bands are shown to originate from strong confinement of the e-h pairs in the cores by these innovative AGS shells, which show a much superior passivating capacity as compared to conventional ZnS shells and recently reported GaS_x shells. Along with ultimate PL QYs, the core/shell QDs show a much higher absorption cross-section when compared to other Cd,Pb,Hg-free QDs, such as ZnSeTe and InP. The combination of the high absorption cross-section, a relatively large Stokes shift and narrow PL bands makes the present AIGS/AGS QDs an ideal candidate for light down-shifting applications. The superiority of the QDs has been convincingly shown by the authors for model luminescent light concentrating films.

The original precursors, QD cores and core/shells are extensively characterized by a well-tailored combination of the physical methods, the quality and reproducibility of the reported results can be evaluated as very high.

The reported results are expected to be of high significance for the field of emissive QDs, because they pave new ways to toxic-metal free emitters and show the feasibility of tailored and narrow-band emission for ternary QDs, which was earlier assumed to be typically only to II-VI and IV-VI QDs. The paper is well written, clearly organized and supported by extensive ESI. No critical comments appeared during the analysis of the manuscript. The paper is performed on a high instrumental level, logically presented, shows high importance and relevance to the target audience, and, therefore, can be recommended for the publication in the present form without revisions.

Reviewer #3 (Remarks to the Author):

The authors prepared AIGS/AGS core/shell quantum dots using a new intermediate complex (Ag-S-Ga(COOR)₂), which was prepared by reacting Ag₂S nanoparticles with excess Ga(OA)₃ and sulfur. The reaction of Ag-S-Ga(COOR)₂ with In(OA)₃ and elemental sulfur at 210 °C produced AIGS core QDs. Then, AGS shells were coated on the preformed AIGS cores by reacting at 240 °C with the same Ag-S-Ga(COOR)₂ and additional sulfur source in the absence of indium sources. STEM-EDS mapping revealed the presence of In-rich AIGS core and AGS shells; the latter is essential for the band-edge PL. The narrowest PL fwhm as an ensemble is 30 nm. Single particle measurements revealed 17 nm fwhm and a longer blink-on time after AGS shell formation. Air-exposure tests recorded half-life periods between 40–135 h depending on the AGS shell thickness. The potential as a material for LSCs was evaluated to show the advantages against InP and ZnSeTe QDs.

First, the developed method produced high-quality quaternary cadmium-free QDs that are worth reporting. However, due to the highly competitive nature of this field, the optical properties of the resulting QDs must be handled carefully. For example, the small difference in the fwhm of PL spectra between 30 nm and 35 nm is significant. Therefore, it is desirable to quantify the spectra of the ensembles in Figure 2e and summarize them in Table S1 (only the data of the 522 nm peak and 30

nm fwhm was provided). Second, I consider that the synthesis of multinary QDs is still challenging, according to the recent review papers. Therefore, more space can be devoted to the synthesis (currently less than 1 page) than the description of LSC experiments, which seems less significant because it was performed in solution (without solidification).

In summary, this paper is considered publishable in Nature Communication, but there is room for revision regarding experimental uncertainty.

Individual comments:

Figure 1a: According to the scheme, AIGS cores were produced by reacting Ag-S-Ga(COOR)₂ complex, In(OA)₃, and S (yellow circle). However, no S or S-OAm was added during core synthesis according to the experimental procedure (page 14, lines 289–295).

Figure 1a: Why do the authors describe the complex as Ag-S-Ga(COOR)₂ in the main text, whereas Ag-S-Ga(OA)₂ in scheme 1a? If these are the same species, please avoid using different chemical equations.

Figure 1b and Experimental section: Conditions for MALDI analysis should be displayed (matrix, cationization reagent, positive or negative modes). The theoretical m/z values of the Ag-S-Ga(OA)₂ are 772.3 (100%), 770.3 (62%), and 774.3 (43%), and it should be higher than these values if ionization is done by proton or other cations (Na⁺ or K⁺ depending on measurement conditions) mixed as a cationization reagent. However, the highest peak in Figure 1b looks m/z = 768. What kind of ions do the authors consider to have been detected?

Page 4, lines 69–72: A recent paper, 10.1021/acs.chemmater.2c03023, seems to achieve 31 nm fwhm and 50–99% PLQY for AIGS-based core/shell QDs. The authors are encouraged to compare the results and methods described in that paper since they took different approaches to overcome the reactivity gap between metal cations.

Page 5: According to the experimental procedure, the Ag:Ga:S composition ratio in the Ag-S-Ga(OA)₂ stock solution is 1:2.25:4.5, indicating an excess of Ga and S compared to the complex. Since the solution was used without isolation, how do these excess species affect the reaction? In particular, during the AGS shell growth, 0.15 mmol sulfur is added to the reaction, which is 10 times the amount of Ag (0.015 mmol). Assuming that 100% of the Ag is reacted form AgGaS₂, where did the excessive gallium and sulfur species go? For example, can the possible reaction of the free gallium with sulfur be ignored?

Page 5, line 107 "Ag-S-Ga(COOR)₂ stock solution is kept at the room temperature for AIGS core synthesis and AGS shell growth." I understand the experimental procedures, but it sounds like AIGS and AGS can be produced at room temperature. I may be corrected, for example, to "kept at room temperature before it was used for AIGS core synthesis..."

Page 6, line 132: Why are ICP results only used for measuring In/Ga ratio? Is it possible to show the data of all elements contained in AIGS/AGS core/shell QDs? It may help understand the structure and homogeneity of QD ensemble. The Ag(In_xGa_{1-x})S₂/AgGaS₂

Table S1: Which spectrum in the main text exhibits 30-nm fwhm? The spectrum of X = 0.5 in Figure 2e looks around fwhm = 35 nm. As the authors compare with the previous data by other groups, these values are crucial in this field. It is recommended that all data in Figure 2e are summarized in Table S1.

Figure S8e Change the X values in the legend from percentages (X = 20, 50, 60...) to ratios (X = 0.2,

0.5, 0.6...) as in the rest of this paper.

Figure S12: Can the authors show the ensemble PL spectra of the AIGS cores? Only the spectrum of the AIGS core with $X=0.5$ is shown in Figure 2d.

Figure 3e: Is there a reason why the time axis is displayed as a logarithm? It should be shown in a linear plot. If the change in luminescence properties is exponential (for some reason) and the authors want to illustrate the half-life period by the shape of the plots, it would be normal to use a logarithmic plot for the vertical axis (PL QY).

Page 9, lines 181–187: Although the high crystallinity and rigid, homogeneous morphology of the AIGS shell is evident in the HRTEM and STEM images, the extension of the PL half-life with increasing shell thickness in air exposure tests is, in my thinking, shorter than expected. Since AgGaS₂ is considered to be air stable as they have been used as photocatalysts, it seems strange that these materials over 2 nm thickness were damaged over 150 hr. In reality, ZnS shell can withstand for years under air. Can the authors rule out the possibility that the shell surface or its exterior (e.g. ligands) is more important for band-edge emission? For example, did the AIGS/AGS core/shell QDs still show band edge PL after repeated purification steps to prepare STEM samples?

Page 15, line 322–324: The author wrote that “LSCs with NC dispersion in toluene were loaded in a clear quartz cuvette (100 × 500 × 10 mm³) to avoid inevitable PL QY drop during the polymerization process”

How did the use of crystal cuvettes avoid a decrease in PL QY? Suppose this statement implies that LSC was evaluated using toluene solutions and liquid containers rather than polymer matrices, it undermines the value of this study since the development of LSC involves difficulties in embedding the QDs into solid materials by polymerization or dissolution in a melt. I would ask the authors to reconsider whether the LSC results are appropriate for inclusion in the main text.

Supplementary note 3: What was the basis function used for the calculation? How was the particle size (quantum size effect) reflected in the calculation? It should be difficult to include all the elements contained in $d = 5\text{--}10$ nm nanocrystals in the calculation. Can the authors provide references to their methodology?

Reviewer #4 (Remarks to the Author):

Authors report the heteroepitaxy for AIGS-AgGaS₂ (AIGS₃₂-AGS) core-shell NCs with near-unity PL QYs in the visible range (460 nm to 620 nm) and their optical/structural properties as well as application in displays and luminescent solar concentrators. The AIGS NCs exhibit the best luminescent performances such as high PL QY, high absorption cross-section and narrow emission linewidth than the NCs reported previously. The experimental results are novel and very interesting. Therefore, this manuscript is recommended for publication in Nature Communications after the following issues are considered carefully.

1. Since high performance AIGS-AGS core-shell NCs were synthesized in this work, authors need to discuss the growth processes of NCs in detail. Especially they should provide clear description and supporting information. Authors think that the key to synthesize high performance AIGS-AGS NCs is the use of Ag-S-Ga(COOR)₂ complex. The growth mechanism of NCs is only supported by NMR. Nuclear magnetic spectroscopy may provide more information on growth kinetics of NCs.
2. The description of the picture information is not exhaustive. The information shown in the picture is not reflected in the text description.

3. Why are the NCs with core radius $r=1.8$ nm used by the author in Fig. 2, while the NCs with core radius $r=1.6$ nm are used to describe the optical properties in Fig. 3? How to select NCs with the $r=1.8$ nm and 1.6 nm? How does the size of NC core affect the optical properties of quantum dots?
4. The component description of quantum dots is not accurate enough. The author uses In content (X) in AIGS core to describe components. X is needed to be described or defined, for example, $X=\text{In}/\text{Ga}$ or $X=\text{In}/(\text{In}+\text{Ga})$ or $X=\text{In}/(\text{Ag}+\text{In}+\text{Ga}+\text{S})$.
5. When authors study the effect of components on PLQY of NCs, they obtain the PL QY of 50% for the NCs with $X=0.2$. The reason of low PLQY is related to the large band gap of cores and the AGS shell coating that can not effectively limit the carriers. It is noted that in Fig. 2f the overall PLQY also shows a downward trend with the increase of X when $X>0.5$. This obviously cannot be explained by the model of shell confined carriers.
6. Most of the reported narrow-band AIGS quantum dots have a long tail in PL bands. In this article, Fig 2e and Fig. S11 show different degrees of tailing in PL bands only when $X=0.2$ and $X=0.9$. Authors should highlight this point and give reasonable explanations.
7. Can the optical properties of AIGS-AGS NCs be improved further if the ZnS shell with a wider gap will continue to be coated?
8. In Fig. 2C, the XRD diffraction pattern of AIGS cores appears to deviate from the standard Tetragonal- AgInS_2 , which should be explained.
9. In Fig. S12, (b,d) should be (b,e) in the caption.

Response to Reviewers

Reviewer #1 (**Publish after revision noted**):

General Comment: This work by Lee *et al.* reports on the synthesis of core shell Ag(In,Ga)S₂/AgGaS₂ nanocrystals for optoelectronic applications. The interest in I-III-VI₂ semiconductors is timely in particular to replace heavy metal-based compounds as light emitters. The chemistry of I-III-VI₂ nanocrystals and in particular the fine control of their surface defects is, however, substantially more complicated than common II-VI or III-V systems resulting in typically broad emission lines and lower emission yields. The authors tackle this challenge through an heteroepitaxy approach that results in type I band alignment, near unity emission yield and narrow emission spectra.

The chemical/photophysical results are interesting but the application perspectives, in particular for luminescent solar concentration (LSCs), do not seem fully justified in particular with respect to the state of the art. The way some data is reported (*e.g.* Fig. 2d) makes it hard to extract the relevant information (absorption/emission overlap integral). Overall, the paper could be suitable for Nature Comm but some amendments are necessary.

Response: We sincerely appreciate Reviewer #1's evaluation (**Publish after revision noted**) and constructive comments and suggestions on our work. All authors thoroughly looked over all comments and suggestions, prepared the responses, and revised the manuscript accordingly. Specifically, per Reviewer #1's comments, we have discussed the comparison of AIGS NCs with CIS NCs in respects to their photophysical mechanism and applicability to LSCs in the revised version of manuscript. Below, we enclose the point-by-point responses to each comment raised by the Reviewer #1. We look forward that the revised version of manuscript sufficiently addresses all the issues and now suitable for publication in *Nature Communications*.

Comment #1-1: The achievement of near unity photoluminescence yield and narrow emission with I-III-VI₂ system is highly relevant also for better understanding the physics of twin compounds such as CuInS₂ or AgInS₂ nanocrystals for which the single particle emission spectrum is still largely debated (see Gamelin vs Klimov results on single CuInS₂ QD spectroscopy). The authors focus on the chemistry and do not discuss their findings in this framework which is actually quite relevant. I strongly suggest to do so and possibly discuss the possible emission mechanism of their systems which is now completely lacking. This would make the paper much more interesting and complete and offer the community a new valuable piece of information. There are also important works on the emission mechanisms in AgInS₂ and CuInS₂ that are not cited and would help to contextualize the results better.

Response: We thank Reviewer #1 for his/her constructive comment that indeed helps enrich scientific discussion in our work. Our present work mainly focuses on the chemistry to attain coherent heteroepitaxy of I-III-VI₂ Ag(In,Ga)₂S₂-AgGaS₂ (AIGS-AGS) core-shell nanocrystals. As the reviewer pointed out, Ag(In,Ga)₂S₂ is a twin compound of CuInS₂ (CIS) NCs, but their emission mechanisms are different. CIS NCs show large Stokes shift and broad PL emission arising from the optical transition between the lowest quantized state for electron and the hole trap state from the multiple oxidation states of Cu (*e.g.*, cuprous (Cu⁺) and cupric (Cu²⁺)) in CIS^{1,2}. By contrast, AIGS-AGS NCs are free from such internal trap states, and thus these NCs emit photons from the optical transition between the lowest quantized states of hole and electron (**Fig. 2d**, **Fig. 2e** and **Fig. 3c**).

We have added the discussion on the comparison of AIGS-AGS NCs with CIS NCs in the revised version of manuscript and cited relevant reports on emission mechanism of CIS NCs.

Change made in the manuscript
(Page 14, line 290-301)

We conclude by noting that the photophysical properties of AIGS-AGS NCs are in stark contrast to those of its twin compound CuInS₂ (CIS) NCs, thereby expanding the application scope of I-III-VI₂ NCs. CIS NCs exhibit high PL QYs in the red to near-IR region with large Stokes shift (300-500 meV), broad emission linewidth (FWHM > 400 meV), and delayed radiative decay time (~200 ns) due to Cu hole trap states^{44,45} from two different oxidation states of Cu, namely cuprous (Cu⁺) and cupric (Cu²⁺). These characteristics guarantee a wide range of solar spectrum absorption and the suppression of energy loss *via* reabsorption processes, which are best adapted for high efficiency LSCs^{20,46-49}. AIGS-AGS NCs, in contrast, exhibit high PL QYs in the visible region with narrow emission linewidth (FWHM ~ 140 meV) and rapid radiative decay time (~40 ns), making them suitable for use in displays, lightings, and colorful LSCs^{24,36-39}. In particular, the LSC with AIGS-AGS NCS will provide aesthetic appealing to energy harvesting system incorporating architectures with small losses^{22,50-52}.

(Page 23, Reference section)

44. Whitham, P. J. et al. Single-particle photoluminescence spectra, blinking, and delayed luminescence of colloidal CuInS₂ nanocrystals. *J. Phys. Chem. C*. **120**, 17136-17142 (2016).
45. Zang, H. et al. Thick-shell CuInS₂/ZnS quantum dots with suppressed “blinking” and narrow single-particle emission line widths. *Nano Lett.* **17**, 1787-1795 (2017).

Comment #1-2: It is impossible to evaluate the spectral overlap between the absorption and PL spectra of shelled crystals from Fig 2d, but based on the position of the excitonic peak for the bare nanocrystals, the spectral overlap seems to be comparable with the exciton emission in III-V nanocrystals. This is interesting from a fundamental perspective but is a technological problem as III-Vs are unsuitable for solar concentration because of excessive self-absorption. The authors should re-plot figure 2 and evaluate the LSC applicability based on quantitative figures (eg. Stokes shift, effective overlap for a given nanocrystal concentration ecc.). There are relevant reports on LSCs that are not cited that would help this analysis.

Response: We normalized PL spectra of **Fig. 2d** to evaluate the LSC applicability of our samples (**Supplementary Fig. 8**). In addition, we compared the overlap of two types of NCs, AIGS-AGS and InP-ZnSeS NCs, in the revised supplementary information based on previous relevant works^{3,4}. The average Q_{LSC} , the ratio between absorption of NCs at the incident light and its emission spectrum, is calculated to be greater for AIGS-AGS NCs than that of reference NCs (InP-ZnSeS). InP-ZnSeS NCs show a smaller Q_{LSC} due to the large overlap between 1S excitonic peak and PL spectra, while the suppressed absorption of AIGS-AGS NCs in their emission spectrum leads to a higher Q_{LSC} . The average Q_{LSC} of AIGS-AGS NCs (18.5) under 420 LED is similar to that of state-of-the-art luminophores studied in the previous paper³. The simulation based on the optical properties of NCs agrees well with the experimental results (**Fig. 4e** and **4f**). Thus, we believe that AIGS-AGS NCs are a prospective luminophore for highly efficient LSCs.

We added detailed explanation on theoretical analysis of LSC applicability in **Supplementary Note 4** and cited relevant reports on LSCs.

Change made in the manuscript

(Page 12, line 250-257)

The optical efficiency (η_{opt}) of LSC, defined by the ratio between incident photons through the top window and collected photons to the edge-mounted solar cells, is normally a function of incident photon absorption, re-emission efficiency of luminophore, re-absorption, and waveguiding efficiency of structure (see **Supplementary Note 4**).⁴⁰⁻⁴² Thus, the desirable characteristics of luminophores are a high absorption coefficient at the wavelength of the incident light, high PL QY, and a large stoke shift. Our AIGS-AGS NCs fulfill the requirements of luminophores for LSCs at the point of absorption and re-emission efficiency (~ 95 %).

(Page 24, Reference section)

46. Velarde, A. s. R. et al. Optimizing the aesthetics of high-performance CuInS₂/ZnS quantum dot luminescent solar concentrator windows. *ACS Applied Energy Materials* **3**, 8159-8163 (2020).
47. Bergren, M. R. et al. High-Performance CuInS₂ Quantum Dot Laminated Glass Luminescent Solar Concentrators for Windows. *ACS Energy Letters* **3**, 520-525 (2018).
48. Sumner, R. et al. Analysis of optical losses in high-efficiency CuInS₂-based nanocrystal luminescent solar concentrators: balancing absorption versus scattering. *J. Phys. Chem. C* **121**, 3252-3260 (2017).
49. Neo, D. C., Goh, W. P., Lau, H. H., Shanmugam, J. & Chen, Y. F. CuInS₂ Quantum Dots with Thick ZnSe x S_{1-x} Shells for a Luminescent Solar Concentrator. *ACS Applied Nano Materials* **3**, 6489-6496 (2020).

Change made in Supplementary Information
(Supplementary Fig. 8)

Supplementary Fig. 8 | 2D plot UV-Vis and PL spectra of AIGS ($X = 0.5$, $r = 1.8$ nm)-AGS NCs with varying shell thicknesses ($0 \leq l \leq 2.9$ nm).

(Page 5)

Supplementary Note 4. Theoretical model for LSC based on optical properties of QDs.

Theoretical optical efficiencies of LSC with AIGS-AGS and InP/ZnSeS NCs were calculated based on previously suggested models^{8,9}. The optical efficiency (η_{opt}) of LSC can be expressed as following:

$$\eta_{opt} = \frac{(1 - R)(1 - e^{-\langle \alpha_1 \rangle})\eta_{PL}\eta_{tr}}{1 + 4\beta G \langle Q_{LSC} \rangle^{-1} (1 - \eta_{PL}\eta_{tr}) \langle \alpha_1 \rangle}$$

Where R is the reflectance of LSC to the incident light, $\langle \alpha_1 \rangle$ is the average absorbance of QDs at incident light, η_{PL} is PL QY of QDs (~ 0.95), η_{tr} is waveguiding efficiency at the interface between quartz glass and air (~ 0.75), β is constant, G is a geometric factor (10) and $\langle Q_{LSC} \rangle$ is the average quality factor. Here, the $\langle Q_{LSC} \rangle$ is the ratio of $\langle \alpha_1 \rangle$ and $\langle \alpha_2 \rangle$, where $\langle \alpha_2 \rangle$ is the average absorbance of QDs under the emission spectrum of QDs, as following.

$$\langle \alpha_1 \rangle = \frac{\int \alpha_1(\lambda)\Phi_{LED}(\lambda)}{\int \Phi_{LED}(\lambda)}$$

$$\langle \alpha_2 \rangle = \frac{\int \alpha_2(\lambda)\Phi_{PL}(\lambda)}{\int \Phi_{PL}(\lambda)}$$

$$\langle Q_{LSC} \rangle = \frac{\langle \alpha_1 \rangle}{\langle \alpha_2 \rangle}$$

where, $\Phi_{LED}(\lambda)$ and $\Phi_{PL}(\lambda)$ are flux density spectrum of excited LED and PL of luminophore. The calculated $\langle Q_{LSC} \rangle$ of AIGS-AGS and InP/ZnSeS NCs are 18.48 and 15.27, respectively. Due to the developed 1s peak of InP/ZnSeS NCs, its $\langle \alpha_2 \rangle$ is higher than that of AIGS-AGS NCs. We believe that improved $\langle Q_{LSC} \rangle$ of AIGS-AGS NCs leads to better η_{opt} in the experiment and theoretical analysis.

(Page 23, Supplementary References section)

- Gungor, K., Du, J. & Klimov, V. I. General Trends in the Performance of Quantum Dot Luminescent Solar Concentrators (LSCs) Revealed Using the “Effective LSC Quality Factor”. *ACS Energy Letters* **2022**, 7(issue), 1741-1749.

Comment #1-3: The comparison with ZnTe and InP nanocrystals for LSCs is irrelevant as neither are suitable LSC emitters. The comparison should be made with AgInS₂ or CuInS₂ systems that represent the state of the art in the field and needs to be done much more rigorously, for example by comparing light propagation for a given solar absorptance.

Response: As the reviewer pointed out, the CuInS₂ (CIS) NCs are one of the best luminophores for LSC due to its large stoke shift and a broad range of light absorption. Moreover, its emission peak can be tuned by controlling the size, enabling us to design special luminophores for various edge-mounted solar cells. Thus, the comparison of AIGS-AGS NCs with state-of-the-art NCs will be helpful for understanding the attainable advantage of LSC with AIGS-AGS NCs. However, it is impossible to empirically compare them because of the substantial disparity in energy gap of these materials: red – NIR emission from CIS NCs *versus* visible emission from AIGS NCs. If the AIGS-AGS NCs extends their absorption spectrum to NIR without deteriorating their PL QY, it will be comparable with CIS NCs. Considering these factors, we theoretically estimate the performance of LSCs implementing AIGS-AGS NCs under various conditions as followings.

1. Experimentally synthesized AIGS-AGS NCs under 420 nm excitation

Based on the Monte Carlo simulation, we derived the loss and gain of the LSC with AIGS-AGS NCs under 420 nm. **Fig. R1** shows optical efficiency and loss in LSC with various concentrated AIGS-AGS NCs. Here we assumed that the size of LSC is $50 \times 50 \times 1 \text{ mm}^3$, similar to a conventional window. Moreover, photovoltaic cell is mounted only one edge of glass ($50 \times 1 \text{ mm}^2$). As the concentration of NC increases, the optical efficiency of LSC increases. It is mainly attributed to improved incident light absorption of LSCs. However, escaped and re-absorbed photons also increase under highly concentrated NCs. Despite the optical loss, the efficiency of LSC with AIGS-AGS NCs reaches around 8% because of its high PL QY.

Fig. R1 | Optical gain and loss in LSC with AIGS-AGS under 420 nm LED irradiation. Here the PV is photons waveguided to edge-mounted photovoltaic cell, the Reabs is loss from re-absorption, the Escape is escaped photons from LSC, and the Non-abs is non-absorbed photons by luminophores.

2. Experimentally synthesized AIGS-AGS NCs under AM 1.5 G irradiation

Empirically synthesized AIGS-AGS NCs exhibit poor absorption of solar light (AM 1.5 G irradiation) as they primarily absorb UV and blue light (**Fig. R2**). Thus, most of the incident solar light penetrates the LSC containing these NCs, resulting in a low theoretical efficiency of only 1 %, which is significantly lower than state-of-art LSCs with CIS.^{5,6} Although the improved absorption coefficient and PL QY of AIGS-AGS NCs, their narrow absorption band results in poor performance of LSC with them.

Fig. R2 | (a) Optical gain and loss in LSCs with experimentally synthesized AIGS-AGS QDs under AM 1.5 G conditions. (b) Magnified view of (a).

3. Ideal AIGS-AGS NCs under AM 1.5 G irradiation

As shown in **Fig. R2**, the broad absorption of incident light is critical in realizing highly efficient LSC. However, it is challenging to synthesize narrow band gap AIGS-AGS NCs at the current status. Hence, we theoretically calculate the optical efficiency of LSC with ideal AIGS-AGS QDs, absorbing a broad range of solar light without deteriorating high PL QYs. We assumed that the PL peak of ideal QDs is around 1000 nm, close to the band gap of conventional c-Si photovoltaic cells. Moreover, the ideal AIGS-AGS QDs exhibit high absorption up to 750 nm and begin to decrease their absorption above 750 nm. In addition, we assumed that the overlap between the absorption and emission spectra of ideal QDs is the same as empirically synthesized QDs, as shown in **Fig. R3**.

Fig. R3 | The PL and absorption spectrum of ideal AIGS-AGS QDs, along with photon flux of solar light. Here, we assumed that the band of AIGS becomes narrow with maintaining a high PL QY (> 0.95).

The optical efficiency of LSC with ideal AIGS-AGS NCs can exceed 7.5 % (**Fig. R4**), comparable with CIS QDs having 91 % of PL QY.⁵ Especially its high PL QY with wide range absorption allows us to utilize highly efficient LSCs. If the further studies about extending the absorption range of QDs are conducted, the highly efficient LSC will be achieved.

Fig. R4 | (a) Optical gain and loss in LSC with ideal AIGS-AGS QDs under AM 1.5 G conditions. (b) Magnified view of (a).

Again, the direct comparison of AIGS NCs and CIS NCs as the luminophores of LSCs is not likely due to the substantial disparity in energy gap of materials. In practice, CIS NCs are best adapted for high efficiency LSCs. By contrast, AIGS-AGS NCs having strong extinction coefficient at UV region and narrow emission at visible region are suitable for use in colorful LSCs that provide aesthetic appealing to energy harvesting system incorporating architectures with small losses.

We have included a comparison of the applicability of AIGS-AGS NCs *versus* CIS NCs to LSCs in the revised manuscript and cited relevant reports.

Change made in the manuscript

(Page 14, line 290-301)

We conclude by noting that the photophysical properties of AIGS-AGS NCs are in stark contrast to those of its twin compound CuInS_2 (CIS) NCs, thereby expanding the application scope of I-III-VI₂ NCs. CIS NCs exhibit high PL QYs in the red to near-IR region with large Stokes shift (300-500 meV), broad emission linewidth (FWHM > 400 meV), and delayed radiative decay time (~200 ns) due to Cu hole trap states^{44,45} from two different oxidation states of Cu, namely cuprous (Cu^+) and cupric (Cu^{2+}). These characteristics guarantee a wide range of solar spectrum absorption and the suppression of energy loss *via* reabsorption processes, which are best adapted for high efficiency LSCs^{20,46-49}. AIGS-AGS NCs, in contrast, exhibit high PL QYs in the visible region with narrow emission linewidth (FWHM ~140 meV) and rapid radiative decay time (~40 ns), making them suitable for use in displays, lightings, and colorful LSCs^{24,36-39}. In particular, the LSC with AIGS-AGS NCS will provide aesthetic appealing to energy harvesting system incorporating architectures with small losses^{20,50-52}.

(Page 24, Reference section)

46. Velarde, A. s. R. et al. Optimizing the aesthetics of high-performance $\text{CuInS}_2/\text{ZnS}$ quantum dot luminescent solar concentrator windows. *ACS Applied Energy Materials* **3**, 8159-8163 (2020).
47. Bergren, M. R. et al. High-Performance CuInS_2 Quantum Dot Laminated Glass Luminescent Solar Concentrators for Windows. *ACS Energy Letters* **3**, 520-525 (2018).
48. Sumner, R. et al. Analysis of optical losses in high-efficiency CuInS_2 -based nanocrystal luminescent solar concentrators: balancing absorption versus scattering. *J. Phys. Chem. C* **121**, 3252-3260 (2017).
49. Neo, D. C., Goh, W. P., Lau, H. H., Shanmugam, J. & Chen, Y. F. CuInS_2 Quantum Dots with Thick $\text{ZnSe}_x\text{S}_{1-x}$ Shells for a Luminescent Solar Concentrator. *ACS Applied Nano Materials* **3**, 6489-6496 (2020).
50. Schliske, S. et al. Design and color flexibility for inkjet-printed perovskite photovoltaics. *ACS Applied Energy Materials* **2**, 764-769 (2018).
51. ter Schiphorst, J. et al. Printed luminescent solar concentrators: Artistic renewable energy. *Energy and Buildings* **207**, 109625 (2020).

52. Bognár, Á. et al. The solar noise barrier project 4: Modeling of full-scale luminescent solar concentrator noise barrier panels. *Renewable Energy* **151**, 1141-1149 (2020).

Comment #1-4: One suggestion for the authors that might add to the appeal of their results is to test the effect of temperature on the emission properties of their nanocrystals. Phonon coupling is a critical aspect in the application of III-V nanocrystals as light converters (both white light and color displays) as it quenches the luminescence and changes the emission color. The core/shell particles reported here could benefit from heterostructuring in both aspects and result substantially better than the state of the art in the field.

Response: As Reviewer #1 suggested, we have conducted temperature dependent PL measurements on AIGS-AGS NCs between RT – 150 °C, at which lightings and displays are generally operated (**Fig. R5**). The characteristics of commercialized InP-ZnSeS NCs having the same optical bandgap are shown for comparison. Both types of NCs show gradual decrease in PL intensities along with spectral shift to lower energies as the temperature increases, and the extents in PL QY drop and spectral shift are similar for both types of NCs in the given temperature variation. From given experimental results, we do not find a clue that AIGS-AGS NCs benefit from heterostructuring (in respect to the phonon coupling) compared to commercialized NC emitters.

Nevertheless, we believe that this information itself must be useful for readers who are interested in use of AIGS-AGS NCs in photonic applications, and thus have added them to the revised version of Supplementary Information.

Fig. R5 | Temperature dependent (a,b) PL spectra of (a) AIGS ($X = 0.5$, $r = 1.8 \text{ nm}$)-AGS ($l = 1.6 \text{ nm}$) and (b) InP-ZnSeS (InP ($r = 1.2 \text{ nm}$)-ZnSe ($l = 1.8 \text{ nm}$)-ZnS ($h = 0.5 \text{ nm}$)) NCs and (c) their color coordinates. The insets of (a) and (b) show temperature dependent PL QYs. The marked lines indicate corresponding temperature.

We made a note in the manuscript and included the experimental results in the Supplementary Information (**Supplementary Fig. 20**) for readers who are interested in use of AIGS-AGS NCs in photonic applications.

Change made in the manuscript

(Page 17, line 362-364)

The temperature dependent PL characteristics of solution NC samples were acquired with EMCCD camera (Princeton Instruments, ProEM HS1024BX3) under excitation with 450 nm lamp (CooLED, pE-300white series SB) (**Supplementary Fig. 20**).

Change made in Supplementary Information

(Supplementary Fig. 20)

Supplementary Fig. 20 | Temperature dependent (a,b) PL spectra of (a) AIGS ($X = 0.5$, $r = 1.8$ nm)-AGS ($l = 1.6$ nm) and (b) InP-ZnSeS (InP ($r = 1.2$ nm)-ZnSe ($l = 1.8$ nm)-ZnS ($h = 0.5$ nm)) NCs and (c) their color coordinates. The insets of (a) and (b) show temperature dependent PL QYs. The marked lines indicate corresponding temperature.

Reviewer #2 (Publish as is):

General Comment: The manuscript reports on highly emissive core/shell quantum dots with Ag-In-Ga-S (AIGS) cores and Ag-Ga-S (AGS) shells produced by using a general and new precursor. A molecular precursor combining Ag, Ga, and S is proposed that allows to equalize the reactivity of Ag, In, and Ga and achieve very homogeneous and controlled Ag-In-Ga-S NC nucleation as well as subsequent formation of a protective Ag-Ga-S shell.

The AIGS/AGS QDs show unprecedentedly high PL QY reaching almost unity along with a very low spectral widths of PL bands, which have never been reported so far for ternary I-III-VI QDs. Single-QD studies showed a high suppression of blinking and non-emissive states in cores due to epitaxial deposition of the shell which has a minimal lattice mismatch with the core. Both ultimate PL QYs and very low FWHMs of PL bands are shown to originate from strong confinement of the e-h pairs in the cores by these innovative AGS shells, which show a much superior passivating capacity as compared to conventional ZnS shells and recently reported GaS_x shells. Along with ultimate PL QYs, the core/shell QDs show a much higher absorption cross-section when compared to other Cd,Pb,Hg-free QDs, such as ZnSeTe and InP. The combination of the high absorption cross-section, a relatively large Stokes shift and narrow PL bands makes the present AIGS/AGS QDs an ideal candidate for light down-shifting applications. The superiority of the QDs has been convincingly shown by the authors for model luminescent light concentrating films.

The original precursors, QD cores and core/shells are extensively characterized by a well-tailored combination of the physical methods, the quality and reproducibility of the reported results can be evaluated as very high.

The reported results are expected to be of high significance for the field of emissive QDs, because they pave new ways to toxic-metal free emitters and show the feasibility of tailored and narrow-band emission for ternary QDs, which was earlier assumed to be typically only to II-VI and IV-VI QDs.

The paper is well written, clearly organized and supported by extensive ESI. No critical comments appeared during the analysis of the manuscript. The paper is performed on a high instrumental level, logically presented, shows high importance and relevance to the target audience, and, therefore, can be recommended for the publication in the present form without revisions.

Response: We are glad to hear that Reviewer #2 recognizes the significance of our work (**Publish as is**). We believe that our findings contribute to broadening the environmentally benign material envelope to the I-III-VI₂ NCs and satisfy broad readership of *Nature Communications*. We thank to the careful consideration of our manuscript for providing us with valuable feedback to improve the quality of our work.

Reviewer #3 (Publish after revision noted):

General Comment: The authors prepared AIGS/AGS core/shell quantum dots using a new intermediate complex Ag–S–Ga(COOR)₂, which was prepared by reacting Ag₂S nanoparticles with excess Ga(OA)₃ and sulfur. The reaction of Ag–S–Ga(COOR)₂ with In(OA)₃ and elemental sulfur at 210 °C produced AIGS core QDs. Then, AGS shells were coated on the preformed AIGS cores by reacting at 240 °C with the same Ag–S–Ga(COOR)₂ and additional sulfur source in the absence of indium sources. STEM-EDS mapping revealed the presence of In-rich AIGS core and AGS shells; the latter is essential for the band-edge PL. The narrowest PL fwhm as an ensemble is 30 nm. Single particle measurements revealed 17 nm fwhm and a longer blink-on time after AGS shell formation. Air-exposure tests recorded half-life periods between 40–135 h depending on the AGS shell thickness. The potential as a material for LSCs was evaluated to show the advantages against InP and ZnSeTe QDs.

First, the developed method produced high-quality quaternary cadmium-free QDs that are worth reporting. However, due to the highly competitive nature of this field, the optical properties of the resulting QDs must be handled carefully. For example, the small difference in the fwhm of PL spectra between 30 nm and 35 nm is significant. Therefore, it is desirable to quantify the spectra of the ensembles in Figure 2e and summarize them in Table S1 (only the data of the 522 nm peak and 30 nm fwhm was provided). Second, I consider that the synthesis of multinary QDs is still challenging, according to the recent review papers. Therefore, more space can be devoted to the synthesis (currently less than 1 page) than the description of LSC experiments, which seems less significant because it was performed in solution (without solidification).

In summary, this paper is considered publishable in Nature Communication, but there is room for revision regarding experimental uncertainty.

Response: We are pleased with Reviewer #3's appraisal (**Publish after revision noted**), constructive comments and suggestions on this work. All authors meticulously reflected all comments and suggestions, prepared the response letter, and revised the manuscript accordingly. Specifically, per Review #3's comments, we have substantially revised the manuscript and Supplementary Information to detail the synthesis procedures and experimental results. Below is the point-by-point responses to the reviewer's comments. We anticipate that the revised version of manuscript is now suitable for publication in *Nature Communications*.

Change made in the manuscript

(Page 5, line 103 - 130)

The role of in-situ generated H₂S in preparation of Ag-S-Ga(OA)₂ is validated by the comparative experiments with *versus* without degassing for H₂S removal (**Supplementary Fig. 3 and Fig. 4**). Specifically, we prepare S-OAm stock solutions degassed at two different temperatures, *i.e.*, room temperature (RT) and 130 °C (at which, the in-situ generated H₂S gas is degassed and collected in cold trap connected to the Schlenk line as yellowish-brown color) (**Supplementary Fig. 3 and Fig. 4**). These S-OAm stock solutions (degassed at RT *versus* 130 °C) are separately added into the reaction flasks containing Ag₂S NPs and the reaction temperature is gradually elevated to 210 °C. Ag₂S NPs mixed with S-OAm degassed at RT are decomposed into a form of AgSH (Ag₂S NPs + H₂S → AgSH, molar ratio of Ag:S = 1:1 is confirmed by ICP-AES), as seen in the gradual decrease in absorbance of Ag₂S NPs and TEM analysis (**Supplementary Fig. 3a and Fig. 5**). In presence of Ga(OA)₃, AgSH simultaneously reacts with Ga(OA)₃ to yield Ag-S-Ga(OA)₂ complex. By contrast, Ag₂S NPs mixed with S-OAm degassed at 130 °C remain unchanged until 170 °C, and decompose into elemental Ag and S at 210 °C (**Supplementary Fig. 3b**). In this case, the present Ga(OA)₃ does not participate in the reaction (**Supplementary Fig. 4**). Above results clearly show that in-situ generated H₂S is the key element to produce Ag-S-Ga(OA)₂ complex.

The formation of Ag-S-Ga(OA)₂ is confirmed with MALDI-TOF mass spectrometry and optical characterization (**Fig. 1b, Supplementary Fig. 5 and Fig. 6**). Ag-S-Ga(OA)₂ stock solution is kept at the room temperature before it was used for AIGS core synthesis and AGS shell growth. The injection of In precursor (In(OA)₃) into Ag-S-Ga(OA)₂ stock solution at an elevated temperature ($T = 210$ °C) bursts AIGS nucleation and subsequent growth. The addition of Ag-S-Ga(OA)₂ stock solution in the AIGS core containing solution allows to grow AGS shell on top of AIGS core. HR-TEM, STEM-HAADF and EDS

analysis verify that resulting NCs are indeed heterostructured in AIGS-AGS core-shell geometry (Fig. 1c-e). It is noted that excess amount of S is deployed throughout the core-shell growth as the S-rich reaction condition guarantees higher PL QY and suppressed trap emission of resulting AIGS-AGS NCs (Supplementary Fig. 7). Reaction procedures for precursor preparations and core-shell growth are detailed in Methods and Supplementary Note 1.

(Page 15, line 312-345)

Ag-S-Ga(OA)₂ preparation. All synthesis was carried out under N₂ atmosphere through the Schlenk line technique. For preparing Ag₂S nanoparticles (NPs), 1 mmol of silver acetate and 5 mL of oleylamine (OAm) were loaded in a three-neck flask and degassed at 50 °C for 1 hr. After the flask was backfilled with N₂, 2.5 ml of dodecanethiol (DDT) and 1 ml of 0.5 M S dissolved in OAm (S-OAm) were injected into the reaction flask to form Ag₂S NPs. To transform Ag₂S NPs to Ag-S-Ga(OA)₂, 4.5 ml of 0.5 M Ga(OA)₃ and 8 ml of 0.5 M S-OAm were added into the reaction flask and temperature was elevated to 210 °C. We note that excess Ga(OA)₃ and S were deployed for complete conversion of Ag₂S NPs to Ag-S-Ga(OA)₂, and the unreacted Ga(OA)₃ and S were used for AIGS core synthesis.

AIGS core synthesis. Reaction flask containing 1.5 ml of Ag-S-Ga(OA)₂ stock solution and 5 ml of OAm is filled with N₂ and heated up to 210 °C. 0.15 ml of 0.5 M In(OA)₃ precursor were swiftly injected into the flask for the nucleation and subsequent growth of AIGS cores (X = 0.5, r = 1.8 nm). We note that an addition of extra S precursor is not needed because excess S-OAm in Ag-S-Ga(OA)₂ stock solution spontaneously participates in the reaction. The reaction temperature was maintained at the elevated temperature for 30 min and cooled to room temperature to cease the reaction. The injected volume of In(OA)₃ precursor was controlled to vary the In content (X) in AIGS cores. Specifically, 0.05, 0.3, 0.6 and 2.7 ml of 0.5 M In(OA)₃ precursor yielded AIGS NCs with In contents (X) of 0.2, 0.6, 0.7 and 0.9, respectively. Resulting AIGS cores were purified twice in glovebox by precipitation (ethanol)/redispersion (toluene) method and finally dispersed in 5 ml of toluene.

AGS shell growth. A reaction flask containing 5 ml of OAm and 300 mg of AIGS cores (X = 0.5) was degassed at 110 °C for 1 hr, back-filled with N₂ and heated up to 240 °C for AGS shell growth. At the elevated temperature, 0.3 ml of Ag-S-Ga(OA)₂ precursor solution, and 0.15 ml of 0.5 M S-OAm were injected, and the reaction temperature was maintained for 1 hr to grow 0.3 nm thick AGS shell. Alternatively, equivalent amount of Ag₂S NPs, Ga(OA)₃ and S-OAm solution were added for AGS shell growth. It is noted that when shell thickness exceeded 0.6 nm, 10 M GaCl₃ solution (dissolved in ethanol, 1 eq of Ga(OA)₃) was added to facilitate uniform shell growth³⁹. For thicker AGS shell growth, Ag, Ga and S precursors were injected repeatedly to grow AGS shell (0.3-0.5 nm for each step) at the fixed reaction temperature (240 °C). Excess amount of S compared to Ag or Ga is deployed throughout the reaction for higher PL QY and suppressed trap emission of AIGS-AGS NCs. Resulting AIGS-AGS NCs were purified twice in glovebox by precipitation (ethanol)/redispersion (toluene) method and finally dispersed in 5 ml of toluene for further characterization and applications.

Change made in Supplementary Information
(Supplementary Fig. 6)

Supplementary Fig. 6 | MALDI-TOF mass spectra of (a) Ga(OA)_3 and (b) Ag-S-Ga(OA)_2 . Dashed red lines in (a) and (b) represent theoretical m/z value from Ag and Ga isotopes. The slight deviation from theoretical m/z value of Ag-S-Ga(OA)_2 is likely originated from Ga(OA)_3 , whose measured values also deviate from theoretical value.

(Supplementary Fig. 7)

Supplementary Fig. 7 | Absorption and PL spectra of AIGS ($X = 0.5$, $r = 1.8 \text{ nm}$)-AGS ($l = 1.6 \text{ nm}$) NCs synthesized from S rich reaction condition (Ag:Ga:S = 1:4.5:10, black line) versus Ga rich reaction condition (Ag:Ga:S = 1:4.5:4.5, red line). The inset shows the magnified view of tail emission in PL spectra.

(Supplementary Fig. 9)

Supplementary Fig. 9 | (a-d) STEM-HAADF image and EDS elemental mapping of Ag (red), In (green), Ga (blue) and S (yellow) for AIGS ($X = 0.5$, $r = 1.8$ nm)-AGS ($l = 1.2$ nm) NCs. (e) Composition ratios of individual AIGS ($X = 0.5$, $r = 1.8$ nm)-AGS ($l = 1.2$ nm) NC marked in Supplementary Fig. 9a. Scale bars are 5 nm.

(Supplementary Fig. 11)

Supplementary Fig. 11 | (a) Absorption, PL spectra and (b) normalized PL spectra of AIGS ($X = 0.5$, $r = 1.8 \text{ nm}$)-AGS ($l = 1.6 \text{ nm}$) NCs as a function of purification steps (precipitation with ethanol and redispersion with toluene). The inset of (b) shows PL QY of AIGS ($X = 0.5$, $r = 1.8 \text{ nm}$)-AGS ($l = 1.6 \text{ nm}$) NCs as a function of purification steps. Addition of TOP into 5-times purified AIGS-AGS NC solution aids to recover PL QY to its initial value.

Comment #3-1: Figure 1a: According to the scheme, AIGS cores were produced by reacting Ag-S-Ga(COOR)₂ complex, In(OA)₃, and S (yellow circle). However, no S or S-OAm was added during core synthesis according to the experimental procedure (page 14, lines 289–295).

Response: The scheme in **Fig. 1a** illustrates the reaction sequences with the key intermediate, *i.e.*, Ag-S-Ga(OA)₂, to form AIGS cores. In practice, excess S-OAm, which is present from the Ag-S-Ga(OA)₂ preparation to ensure 100 % conversion of Ag₂S NPs to AgSH, participates in the nucleation and subsequent growth of AIGS cores. To avoid unwanted misleading, we have revised the Methods section as below.

Change made in the manuscript

(Page 15, line 312-332)

Ag-S-Ga(OA)₂ preparation. All synthesis was carried out under N₂ atmosphere through the Schlenk line technique. For preparing Ag₂S nanoparticles (NPs), 1 mmol of silver acetate and 5 mL of oleylamine (OAm) were loaded in a three-neck flask and degassed at 50 °C for 1 hr. After the flask was backfilled with N₂, 2.5 ml of dodecanethiol (DDT) and 1 ml of 0.5 M S dissolved in OAm (S-OAm) were injected into the reaction flask to form Ag₂S NPs. To transform Ag₂S NPs to Ag-S-Ga(OA)₂, 4.5 ml of 0.5 M Ga(OA)₃ and 8 ml of 0.5 M S-OAm were added into the reaction flask and temperature was elevated to 210 °C. **We note that excess Ga(OA)₃ and S were deployed for complete conversion of Ag₂S NPs to Ag-S-Ga(OA)₂, and the unreacted Ga(OA)₃ and S were used for AIGS core synthesis.**

AIGS core synthesis. Reaction flask containing 1.5 ml of Ag-S-Ga(OA)₂ stock solution and 5 ml of OAm is filled with N₂ and heated up to 210 °C. 0.15 ml of 0.5 M In(OA)₃ precursor were swiftly injected into the flask for the nucleation and subsequent growth of AIGS cores (X = 0.5, r = 1.8 nm). **We note that an addition of extra S precursor is not needed because excess S-OAm in Ag-S-Ga(OA)₂ stock solution spontaneously participates in the reaction.** The reaction temperature was maintained at the elevated temperature for 30 min and cooled to room temperature to cease the reaction. The injected volume of In(OA)₃ precursor was controlled to vary the In content (X) in AIGS cores. Specifically, 0.05, 0.3, 0.6 and 2.7 ml of 0.5 M In(OA)₃ precursor yielded AIGS NCs with In contents (X) of 0.2, 0.6, 0.7 and 0.9, respectively. Resulting AIGS cores were purified twice in glovebox by precipitation (ethanol)/redispersion (toluene) method and finally dispersed in 5 ml of toluene.

Comment #3-2: Figure 1a: Why do the authors describe the complex as Ag–S–Ga(COOR)₂ in the main text, whereas Ag–S–Ga(OA)₂ in scheme 1a? If these are the same species, please avoid using different chemical equations.

Response: To avoid unwanted misleading, we have changed Ag-S-(COOR)₂ to Ag–S–Ga(OA)₂ (highlighted with yellow background) throughout the manuscript and supplementary information.

List of changes made in the manuscript

(Page 2, line 34)

(Page 3, line 60)

(Page 4, line 76-78)

(Page 5, line 97-98, 102)

(Page 14, line 304)

(Page 15, line 313, 318)

List of changes made in the Supplementary Information

(Supplementary Fig. 5b, c and e)

Comment #3-3: Figure 1b and Experimental section: Conditions for MALDI analysis should be displayed (matrix, cationization reagent, positive or negative modes). The theoretical m/z values of the Ag-S-Ga(OA)₂ are 772.3 (100%), 770.3 (62%), and 774.3 (43%), and it should be higher than these values if ionization is done by proton or other cations (Na⁺ or K⁺ depending on measurement conditions) mixed as a cationization reagent. However, the highest peak in Figure 1b looks $m/z = 768$. What kind of ions do the authors consider to have been detected?

Response: We conducted MALDI-TOF mass spectroscopy with Voyager DE-STR (Applied Biosystems). The analyte and matrix (dithranol) were mixed in chloroform without cationization reagent. The dried samples were ionized by pulsed N₂ laser (337 nm, 3 ns pulses). 20.0 kV was applied for accelerating positive ions. The reflector mode was used for MS analysis.

As the reviewer pointed out, we found that the m/z value for Ag-S-Ga(OA)₂ + H⁺ ($m/z = 768$) is measured to be smaller than the theoretical value (Ag-S-Ga(OA)₂ + H⁺ adduct : $m/z = 773.3$) in the chosen measurement condition. We found that this deviation comes from Ga(OA)₃, whose measured m/z value (peak at 911.9) is also measured to be smaller than theoretical value (Ga(OA)₃ + H⁺ adduct : 913.7) (see **Fig. R6a**). The reason for the discrepancies seen in Ga(OA)₃ or Ag-S-Ga(OA)₂ (not for In(OA)₃) is currently undetermined. Nevertheless, this tendency seen in MALDI-TOF together with complementary experiments (*Page 5, line 95-130 in the manuscript* and *Supplementary Fig. 3-5*) suggest that the reaction intermediate is indeed Ag-S-Ga(OA)₂.

We have included the MALDI-TOF results of Ga(OA)₃ and Ag-S-Ga(OA)₂, and specific characterization condition in the revised version of manuscript and Supplementary Information.

Fig. R6 | MALDI-TOF mass spectra of (a) Ga(OA)₃ and (b) Ag-S-Ga(OA)₂. Dashed red lines in (a) and (b) represent theoretical m/z value from Ag and Ga isotopes.

Change made in the manuscript
(Page 17, line 364-368)

MALDI-TOF mass spectroscopy was conducted with Voyager DE-STR (Applied Biosystems). The analyte and matrix (dithranol) were mixed in chloroform. The dried samples were ionized by pulsed N₂ laser (337 nm, 3 ns pulses). 20.0 kV was applied for accelerating positive ions. The reflector mode was used for MS analysis.

Change made in Supplementary Information.
(Supplementary Fig. 6)

Supplementary Fig. 6 | MALDI-TOF mass spectra of (a) Ga(OA)_3 and (b) Ag-S-Ga(OA)_2 . Dashed red lines in (a) and (b) represent theoretical m/z value from Ag and Ga isotopes. The slight deviation from theoretical m/z value of Ag-S-Ga(OA)_2 is likely originated from Ga(OA)_3 , whose measured values also deviate from theoretical value.

Comment #3-4: Page 4, lines 69–72: A recent paper, 10.1021/acs.chemmater.2c03023, seems to achieve 31 nm fwhm and 50–99% PLQY for AIGS-based core/shell QDs. The authors are encouraged to compare the results and methods described in that paper since they took different approaches to overcome the reactivity gap between metal cations.

Response: We have cited the paper in our manuscript and compared the results and methods.

Change made in the manuscript

(Page 4, line 70-72)

The up-to-date chemistry still faces hurdles in attaining homogeneity in AIGS alloyed core synthesis and passivating the surface trap states, as seen from their mediocre optical characteristics (see **Supplementary Fig. 1** and **Supplementary Table 1**).^{34-37, 39}

(Page 10, line 213-215)

Amorphous GaSx shell eliminates the broadband emission of AIGS cores, but only marginal PL QY enhancement (~70 %) is allowed due to its inherent structural imperfection. Higher PL QY is achievable with an aid of delicate surface passivation with ligands³⁹.

(Page 23, Reference section)

39. Uematsu, T., Tepakidareekul, M., Hirano, T., Torimoto, T. & Kuwabata, S. Facile High-Yield Synthesis of Ag–In–Ga–S Quaternary Quantum Dots and Coating with Gallium Sulfide Shells for Narrow Band-Edge Emission. *Chem. Mater.* **35**, 1094-1106 (2023).

Change made in Supplementary Information

(Supplementary Fig. 1)

Supplementary Fig. 1 | PL QY and FWHM comparison among heavy metal-free QDs. Detailed data are listed in **Supplementary Table 1**.

(Supplementary Table 1)

Supplementary Table 1 | Optical properties of heavy metal-free QDs in previous literatures and this work.

	PL peak (nm)	FWHM (nm)	PL QY (%)	Ref.	
InP-ZnSeS	528	36	95	[25]	
	525	38	88	[26]	
	531	34	68	[26]	
	518	42	78	[27]	
	527	36	97	[28]	
ZnSeTe-ZnSeS	524	41	95	[29]	
	520	45	80	[30]	
	500	43	88	[31]	
AIGS-ZnS	544	137	68	[34]	
	618	100	79	[35]	
AIGS-GaSx	518	36	68	[36]	
	568	40	71	[37]	
	528	31	53	[39]	
	543	37	99	[39]	
This work					
	X / r (nm) / l (nm) *	PL peak (nm)	FWHM (nm)	PL QY (%)	Ref.
AIGS-AGS	0.5 / 1.8 / 0.6	518	55	77	Fig. 2d
	0.5 / 1.8 / 1.2	517	34	91	Fig. 2d
	0.5 / 1.8 / 1.6	517	31	96	Fig. 2d
	0.5 / 1.8 / 2.9	517	30	96	Fig. 2d
	0.2 / 1.8 / 1.6	468	34	49	Fig. 2e
	0.6 / 2.2 / 1.6	551	38	95	Fig. 2e
	0.7 / 2.3 / 1.6	565	39	92	Fig. 2e
	0.9 / 2.9 / 1.6	610	42	86	Fig. 2e

* X / r / l = In ratio in AIGS core {[In]/([In]+[Ga])} / AIGS core radius / AGS shell thickness

Comment #3-5: Page 5: According to the experimental procedure, the Ag:Ga:S composition ratio in the Ag-S-Ga(OA)₂ stock solution is 1:2.25:4.5, indicating an excess of Ga and S compared to the complex. Since the solution was used without isolation, how do these excess species affect the reaction? In particular, during the AGS shell growth, 0.15 mmol sulfur is added to the reaction, which is 10 times the amount of Ag (0.015 mmol). Assuming that 100% of the Ag is reacted form AgGaS₂, where did the excessive gallium and sulfur species go? For example, can the possible reaction of the free gallium with sulfur be ignored?

Response: High ratios of Ga(OA)₃ and S-OAm are intentional to ensure 100% conversion of Ag₂S NPs to Ag-S-Ga(OA)₂. Due to the low reactivity of Ga(OA)₃, the extra precursors do not entail unwanted side reactions, such as the homogeneous nucleation of GaS. The excess S-OAm from the preparation of Ag-S-Ga(OA)₂ stock solution participate in the nucleation and growth of AIGS cores. We keep the S-OAm content higher than that of Ga(OA)₃ throughout the growth, because the S-rich condition aids to suppress a tail emission in PL and improve PL QY (from 81 % to 93 %) than the counter case (**Fig. R7**). We speculate that sulfur rich surface has less trap states than gallium rich case. The remaining S and Ga(OA)₃ after the NC synthesis can be easily removed through the standard purification processes (precipitation/redispersion).

Fig. R7 | Absorption and PL spectra of AIGS ($X = 0.5$, $r = 1.8 \text{ nm}$)-AGS ($l = 1.6 \text{ nm}$) NCs synthesized from S rich reaction condition (Ag:Ga:S = 1:4.5:10, black line) versus Ga rich reaction condition (Ag:Ga:S = 1:4.5:4.5, red line). The inset shows the magnified view of tail emission in PL spectra.

To help readers understanding, we have added a note that explains the reason for the presence of excess Ga and S precursors in the **Method** section.

Changes made in the manuscript

(Page 6, line 119 - 130)

The formation of Ag-S-Ga(OA)₂ is confirmed with MALDI-TOF mass spectrometry and optical characterization (**Fig. 1b**, **Supplementary Fig. 5** and **Fig. 6**). Ag-S-Ga(OA)₂ stock solution is kept at the room temperature before it was used for AIGS core synthesis and AGS shell growth. The injection of In precursor (In(OA)₃) into Ag-S-Ga(OA)₂ stock solution at an elevated temperature ($T = 210 \text{ }^\circ\text{C}$) bursts AIGS nucleation and subsequent growth. The addition of Ag-S-Ga(OA)₂ stock solution in the AIGS core containing solution allows to grow AGS shell on top of AIGS core. HR-TEM, STEM-HAADF and EDS analysis verify that resulting NCs are indeed heterostructured in AIGS-AGS core-shell geometry (**Fig. 1c-e**). It is noted that excess amount of S is deployed throughout the core-shell growth as the S-rich reaction condition guarantees higher PL QY and suppressed trap emission of resulting AIGS-AGS NCs (**Supplementary Fig. 7**). Reaction procedures for precursor preparations and core-shell growth are detailed in **Methods** and **Supplementary Note 1**.

(Page 15, line 312-345)

Ag-S-Ga(OA)₂ preparation. All synthesis was carried out under N₂ atmosphere through the Schlenk line technique. For preparing Ag₂S nanoparticles (NPs), 1 mmol of silver acetate and 5 mL of oleylamine (OAm) were loaded in a three-neck flask and degassed at 50 °C for 1 hr. After the flask was backfilled with N₂, 2.5 ml of dodecanethiol (DDT) and 1 ml of 0.5 M S dissolved in OAm (S-OAm) were injected into the

reaction flask to form Ag₂S NPs. To transform Ag₂S NPs to Ag-S-Ga(OA)₂, 4.5 ml of 0.5 M Ga(OA)₃ and 8 ml of 0.5 M S-OAm were added into the reaction flask and temperature was elevated to 210 °C. We note that excess Ga(OA)₃ and S were deployed for complete conversion of Ag₂S NPs to Ag-S-Ga(OA)₂, and the unreacted Ga(OA)₃ and S were used for AIGS core synthesis.

AIGS core synthesis. Reaction flask containing 1.5 ml of Ag-S-Ga(OA)₂ stock solution and 5 ml of OAm is filled with N₂ and heated up to 210 °C. 0.15 ml of 0.5 M In(OA)₃ precursor were swiftly injected into the flask for the nucleation and subsequent growth of AIGS cores (X = 0.5, r = 1.8 nm). We note that an addition of extra S precursor is not needed because excess S-OAm in Ag-S-Ga(OA)₂ stock solution spontaneously participates in the reaction. The reaction temperature was maintained at the elevated temperature for 30 min and cooled to room temperature to cease the reaction. The injected volume of In(OA)₃ precursor was controlled to vary the In content (X) in AIGS cores. Specifically, 0.05, 0.3, 0.6 and 2.7 ml of 0.5 M In(OA)₃ precursor yielded AIGS NCs with In contents (X) of 0.2, 0.6, 0.7 and 0.9, respectively. Resulting AIGS cores were purified twice in glovebox by precipitation (ethanol)/redispersion (toluene) method and finally dispersed in 5 ml of toluene.

AGS shell growth. A reaction flask containing 5 ml of OAm and 300 mg of AIGS cores (X = 0.5) was degassed at 110 °C for 1 hr, back-filled with N₂ and heated up to 240 °C for AGS shell growth. At the elevated temperature, 0.3 ml of Ag-S-Ga(OA)₂ precursor solution, and 0.15 ml of 0.5 M S-OAm were injected, and the reaction temperature was maintained for 1 hr to grow 0.3 nm thick AGS shell. Alternatively, equivalent amount of Ag₂S NPs, Ga(OA)₃ and S-OAm solution were added for AGS shell growth. It is noted that when shell thickness exceeded 0.6 nm, 10 M GaCl₃ solution (dissolved in ethanol, 1 eq of Ga(OA)₃) was added to facilitate uniform shell growth³⁹. For thicker AGS shell growth, Ag, Ga and S precursors were injected repeatedly to grow AGS shell (0.3-0.5 nm for each step) at the fixed reaction temperature (240 °C). Excess amount of S compared to Ag or Ga is deployed throughout the reaction for higher PL QY and suppressed trap emission of AIGS-AGS NCs. Resulting AIGS-AGS NCs were purified twice in glovebox by precipitation (ethanol)/redispersion (toluene) method and finally dispersed in 5 ml of toluene for further characterization and applications.

Change made in Supplementary Information
(Supplementary Fig. 7)

Supplementary Fig. 7 | Absorption and PL spectra of AIGS (X = 0.5, r = 1.8 nm)-AGS (l = 1.6 nm) NCs synthesized from S rich reaction condition (Ag:Ga:S = 1:4.5:10, black line) versus Ga rich reaction condition (Ag:Ga:S = 1:4.5:4.5, red line). The inset shows the magnified view of tail emission in PL spectra.

Comment #3-6: Page 5, line 107 “Ag-S-Ga(COOR)₂ stock solution is kept at the room temperature for AIGS core synthesis and AGS shell growth.” I understand the experimental procedures, but it sounds like AIGS and AGS can be produced at room temperature. I may be corrected, for example, to “kept at room temperature before it was used for AIGS core synthesis...”

Response: Per Reviewer #3’s comments, we have revised the manuscript as below.

Change made in the manuscript

(Page 6, line 120-121)

Ag-S-Ga(COOR)₂ stock solution is kept at the room temperature **before it was used** for AIGS core synthesis and AGS shell growth.

Comment #3-7: Page 6, line 132: Why are ICP results only used for measuring In/Ga ratio? Is it possible to show the data of all elements contained in AIGS/AGS core/shell QDs? It may help understand the structure and homogeneity of QD ensemble. The $\text{Ag}(\text{In}_x\text{Ga}_{1-x})\text{S}_2/\text{AgGaS}_2$

Response: We sincerely appreciate reviewer’s constructive suggestion on our work. It seems that the reviewer concerns the homogeneity among AIGS-AGS NCs. To corroborate homogeneity of quaternary NCs, we have conducted EDS mapping on multiple of individual AIGS-AGS NC and confirmed their compositional homogeneity (see **Fig. R8**). The narrow spectral linewidth of ensemble AIGS-AGS NCs together with small spectral deviation seen among individual AIGS-AGS NC also support the structural and compositional homogeneity of synthesized NCs. We have included the experimental results in Supplementary Information.

Fig. R8 | (a-d) STEM-HAADF image and EDS elemental mapping of Ag (red), In (green), Ga (blue) and S (yellow) for AIGS ($X = 0.5$, $r = 1.8 \text{ nm}$)-AGS ($l = 1.2 \text{ nm}$) NCs. (e) Composition ratios of individual AIGS ($X = 0.5$, $r = 1.8 \text{ nm}$)-AGS ($l = 1.2 \text{ nm}$) NC marked in **Fig. R8a**. Scale bars are 5 nm.

Change made in Supplementary Information

(Supplementary Fig. 9)

Supplementary Fig. 9 | (a-d) STEM-HAADF image and EDS elemental mapping of Ag (red), In (green), Ga (blue) and S (yellow) for AIGS ($X = 0.5$, $r = 1.8$ nm)-AGS ($l = 1.2$ nm) NCs. (e) Composition ratios of individual AIGS ($X = 0.5$, $r = 1.8$ nm)-AGS ($l = 1.2$ nm) NC marked in Supplementary Fig. 9a. Scale bars are 5 nm.

Comment #3-8: Table S1: Which spectrum in the main text exhibits 30-nm fwhm? The spectrum of $X = 0.5$ in Figure 2e looks around fwhm = 35 nm. As the authors compare with the previous data by other groups, these values are crucial in this field. It is recommended that all data in Figure 2e are summarized in Table S1.

Response: We updated **Fig. 2d** and **Fig. 2e** with most optimized AIGS-AGS NCs samples. AIGS ($X = 0.5$, $r = 1.8$ nm)-AGS ($l = 2.9$ nm) NCs in **Fig. 2d** exhibit FWHM of 30 nm. We provide the summary of key optical characteristics of AIGS-AGS NCs from the figures in Supplementary Information.

Change made in the manuscript

(Page 27, Fig. 2d and Fig. 2e)

Fig. 2 | AIGS-AGS NCs with variable core compositions and shell dimensions. (a) Schematic illustrations of the geometry (left) and potential profile (right) of AIGS-AGS NCs. Blue and red lines indicate the lowest quantized energy state for electron ($1S_e$) and hole ($1S_h$), respectively. (b) TEM images (scale bars = 10 nm), (c) X-ray diffraction patterns, and (d) UV-Vis absorption and PL spectra of AIGS ($X = 0.5$, $r = 1.8$ nm)-AGS NCs with varying shell thicknesses ($0 \leq l \leq 2.9$ nm). The inset displays PL QYs as a function of shell thickness. 2d plots of PL spectra are supported in **Supplementary Fig. 8** (e) PL spectra, (f) energy gaps ($1S_e-1S_h$) obtained from UV-Vis spectra (top) and PL QYs (bottom), and (g) their color coordinates of AIGS-AGS ($l = 1.6$ nm) NCs with varying In contents ($0.2 \leq X \leq 0.9$) (inset: a photographic image of AIGS-AGS NCs with varying In ratios ($X = 0.2, 0.3, 0.5, 0.7$ and 0.9 from the left)). Gray triangles in (f) represent our calculation results for which homogeneously alloyed AIGS cores are taken into account. Synthesis and experimental results for AIGS-AGS NCs with varying compositions and shell dimensions are detailed in **Methods, Supplementary Note 1** and **Supplementary Fig. 7-16**.

Change made in Supplementary Information

(Supplementary Table 1)

Supplementary Table 1 | Optical properties of heavy metal-free QDs in previous literatures and **this work**.

	PL peak (nm)	FWHM (nm)	PL QY (%)	Ref.
InP-ZnSeS	528	36	95	[25]
	525	38	88	[26]
	531	34	68	[26]
	518	42	78	[27]
	527	36	97	[28]
ZnSeTe-ZnSeS	524	41	95	[29]
	520	45	80	[30]
	500	43	88	[31]
AIGS-ZnS	544	137	68	[34]
	618	100	79	[35]
AIGS-GaSx	518	36	68	[36]
	568	40	71	[37]
	528	31	53	[39]
	543	37	99	[39]

This work

	X / r (nm) / l (nm) *	PL peak (nm)	FWHM (nm)	PL QY (%)	Ref.
AIGS-AGS	0.5 / 1.8 / 0.6	518	55	77	Fig. 2d
	0.5 / 1.8 / 1.2	517	34	91	Fig. 2d
	0.5 / 1.8 / 1.6	517	31	96	Fig. 2d
	0.5 / 1.8 / 2.9	517	30	96	Fig. 2d
	0.2 / 1.8 / 1.6	468	34	49	Fig. 2e
	0.6 / 2.2 / 1.6	551	38	95	Fig. 2e
	0.7 / 2.3 / 1.6	565	39	92	Fig. 2e
	0.9 / 2.9 / 1.6	610	42	86	Fig. 2e

* X / r / l = In ratio in AIGS core {[In]/([In]+[Ga])} / AIGS core radius / AGS shell thickness

Comment #3-9 & #3-10: Figure S13e Change the X values in the legend from percentages (X = 20, 50, 60...) to ratios (X = 0.2, 0.5, 0.6...) as in the rest of this paper. Can the authors show the ensemble PL spectra of the AIGS cores? Only the spectrum of the AIGS core with X=0.5 is shown in Figure 2d.

Response: We have fixed the X values in **Supplementary Fig. 13e** to be consistent with the rest of the manuscript. Also, we added ensemble PL spectra of AIGS cores with varying In ratio (X) in **Supplementary Fig. 13e**.

Change made in Supplementary Information
(Supplementary Fig. 13e)

Supplementary Fig. 13 | (a-d) TEM images (scale bars = 10 nm), (e) absorption and PL spectra and (f) X-ray diffraction patterns of AIGS cores with varying In ratios ($0.2 \leq X \leq 0.9$). Spectra in (e) and XRD patterns of (f) are vertically shifted for visual clarity.

Comment #3-11: Figure 3e: Is there a reason why the time axis is displayed as a logarithm? It should be shown in a linear plot. If the change in luminescence properties is exponential (for some reason) and the authors want to illustrate the half-life period by the shape of the plots, it would be normal to use a logarithmic plot for the vertical axis (PL QY).

Response: We chose to display the time axis as a logarithm to make a space for an inset in **Fig. 3e** that displays the trend in half-life period (T_{50}) as a function of the shell thickness in AIGS-AGS NCs along with the legends for graphs. We decide to keep this after consideration of readability of the figure.

Comment #3-12: Page 9, line 181–187: Although the high crystallinity and rigid, homogeneous morphology of the AIGS shell is evident in the HRTEM and STEM images, the extension of the PL half-life with increasing shell thickness in air exposure tests is, in my thinking, shorter than expected. Since AgGaS₂ is considered to be air stable as they have been used as photocatalysts, it seems strange that these materials over 2 nm thickness were damaged over 150 hr. In reality, ZnS shell can withstand for years under air. Can the authors rule out the possibility that the shell surface or its exterior (e.g. ligands) is more important for band-edge emission? For example, did the AIGS/AGS core/shell QDs still show band edge PL after repeated purification steps to prepare STEM samples?

Response: We are grateful to Reviewer #3 for his/her constructive comment on our work. As the reviewer pointed out, the control of their surface is the key for the PL QY and stability of nanocrystal emitters. It is known that the broken bonds of the surface atoms of NCs play as the trap states that are responsible for the reduction of PL QYs and photochemical reactions at surfaces (*e.g.*, oxidation). Passivation of NC surfaces with inorganic shells and/or ligands is the representative means to decouple photoexcited charge carriers from the surface trap states, leading to the improvement of the luminescence efficiency and stability of NCs.

In the present study, we devise the coherent heteroepitaxy on AIGS cores with the AGS shell. Resulting AIGS-AGS NCs show near-unity PL QYs, indicating that our strategy for heteroepitaxy does not entail interfacial defects. In addition, AIGS-AGS NCs with greater AGS shell thickness show enhanced stability against oxidation, implying that the AGS shell effectively confines photoexcited charge carriers within the AIGS core. However, the PL QY drop seen in the oxidation test indicate that the surface characteristics is subject to change upon exposure to oxidative conditions. We consider that, as generally accepted for NC emitters, the ligand detachment from the AGS surface and/or oxidation of AGS surfaces are the main processes that create the surface trap states and thus deteriorate photophysical characteristics of AIGS-AGS NCs.

Reviewer #3 questions, among the inorganic shell or the ligand, which one is more effective for the band edge emission of NCs. In principle, any means that are effective to passivate the surface states aid to enhance the efficiency and stability. In case of AIGS NCs, comprehensive understanding on the surface properties is lacking and thus the means to control them with ligands yet to be developed. In fact, the band-edge emission only has not been achieved for AIGS core samples. By contrast, AIGS-AGS NCs of the present study show strong band-edge emission, suggesting the effectiveness of the present approach over the shell surface or ligands.

Reviewer #3 also questions AIGS-AGS NCs show band-edge PL emission after repeated purification steps. **Fig. R9** displays the changes in PL QY of AIGS-AGS NCs as a function of purification steps. PL QYs of AIGS-AGS NCs continue to decrease along the repetition of purification (precipitation with ethanol and redispersion with toluene). The PL QY of AIGS-AGS NCs recovers to the initial value with an addition of trioctylphosphine (TOP). These experiments signify that the native ligands from the chemistry are detached from the surfaces of AIGS-AGS NCs by leaving the broken bonds of surface atoms, resulting in the PL QY drops. However, NCs keep their PL spectra shape originating from the band-edge emission and higher PL QYs (> 50 %) compared to the core only case, apparently suggesting that the AGS shell is more important than the shell surface or ligands in respects to the band-edge emission and PL QY.

Besides, we note that the stability of NCs is not a characteristic that solely depends on the composition of shell materials. For examples, red- or green-emitting CdSe/ZnS NCs are stable over a year, but blue-emitting ZnSeTe/ZnS NCs or CdZnS/ZnS NCs having the same shell and ligands are not stable. The potential profile and electronic energy levels and their spatial distributions in it should be taken into account to understand the stability of NCs.

We hope that our explanation adequately addresses Reviewer #3's comment.

Fig. R9 | (a) Absorption, PL spectra and (b) normalized PL spectra of AIGS ($X = 0.5$, $r = 1.8 \text{ nm}$)-AGS ($l = 1.6 \text{ nm}$) NCs as a function of purification steps (precipitation with ethanol and redispersion with toluene). The inset of (b) shows PL QY of AIGS ($X = 0.5$, $r = 1.8 \text{ nm}$)-AGS ($l = 1.6 \text{ nm}$) NCs as a function of purification steps. Addition of TOP into 5-times purified AIGS-AGS NC solution aids to recover PL QY to its initial value.

We have included the experimental results in Supplementary Information

Change made in Supplementary Information

(Supplementary Fig. 11)

Supplementary Fig. 11 | (a) Absorption, PL spectra and (b) normalized PL spectra of AIGS ($X = 0.5$, $r = 1.8 \text{ nm}$)-AGS ($l = 1.6 \text{ nm}$) NCs as a function of purification steps (precipitation with ethanol and redispersion with toluene). The inset of (b) shows PL QY of AIGS ($X = 0.5$, $r = 1.8 \text{ nm}$)-AGS ($l = 1.6 \text{ nm}$) NCs as a function of purification steps. Addition of TOP into 5-times purified AIGS-AGS NC solution aids to recover PL QY to its initial value.

Comment #3-13: Page 15, line 322–324: The author wrote that “LSCs with NC dispersion in toluene were loaded in a clear quartz cuvette (100 × 500 × 10 mm³) to avoid inevitable PL QY drop during the polymerization process”

How did the use of crystal cuvettes avoid a decrease in PL QY? Suppose this statement implies that LSC was evaluated using toluene solutions and liquid containers rather than polymer matrices, it undermines the value of this study since the development of LSC involves difficulties in embedding the QDs into solid materials by polymerization or dissolution in a melt. I would ask the authors to reconsider whether the LSC results are appropriate for inclusion in the main text.

Response: We agree with Reviewer #3’s point that actual development of LCS involves difficulties in embedding the QDs into solid materials by polymerization. As seen in see **Comment #3-12**, PL QY drops during the repeated purification steps due to the ligand detachment and recovers with an addition of extra phosphine-based ligand, calling for the development of phosphine-based cross-linkable organic molecules to passivate NC surfaces and to disperse NCs into the polymer resins. However, the phosphine groups could deactivate the catalysts that activate polymerization of the resin, and thus the use of the phosphine-based cross-linkable organic molecules necessitates new polymerization chemistry with different types of catalysts, which are beyond the scope of present work.

We have exemplified the use of AIGS-AGS NCs in LSCs in comparison to the commercialized heavy metal-free InP-ZnSeS NCs. We deploy LSCs with NC dispersion in toluene loaded in a cuvette to exclude unwanted effects of NC aggregation or degradation (PL QY drop) during the polymerization processes. Reviewer #3 ponders the effectiveness of our LSC experiment set-up as it does not comprehend the implement of NCs in films. However, we believe that, despite its lack of completeness for their practical use, our LSC set-up is still effective to highlight the figure-of-merits of AIGS-AGS NCs, *i.e.*, high absorption cross-section, suppressed energy transfer and near-unity PL QY, compared to InP-ZnSeS NCs. In fact, these approaches have been widely accepted in previous works^{5,7-9} on LSCs with newly developed materials to open their full potential as luminophores. Therefore, we decide to keep the LSC data in the main text.

Comment #3-14: Supplementary note 3: What was the basis function used for the calculation? How was the particle size (quantum size effect) reflected in the calculation? It should be difficult to include all the elements contained in $d = 5\text{--}10$ nm nanocrystals in the calculation. Can the authors provide references to their methodology?

Response: The exciton energy of the charged QD is calculated by custom written python code where two band K dot P model is implemented^{1,2}. The finite difference scheme converts $K \cdot P$ matrix calculation to linear algebra with equally spaced 1.5\AA mesh. We applied cylindrical symmetry for Laplacian operator so that full 3d simulation is achieved. To be a self-consistency, Schrödinger and Poisson equations are solved iteratively until the energy difference (ΔE) is less than 1 meV. The compositions and radius of AIGS cores experimentally obtained from ICP-AES and TEM (**Supplementary Fig. 14**) are taken into account for the exciton energy calculation. We detailed the calculation method and references in the revised Supplementary Information.

Change made in Supplementary Information

(Page 3, line 44-51)

Energy gap calculation. The exciton energy of the charged QD is calculated by custom written python code where two band K dot P model is implemented^{1,2}. The finite difference scheme converts $K \cdot P$ matrix calculation to linear algebra with equally spaced 1.5\AA mesh. We applied cylindrical symmetry for Laplacian operator so that full 3d simulation is achieved. To be a self-consistency, Schrödinger and Poisson equations are solved iteratively until the energy difference (ΔE) is less than 1 meV. The compositions and radius of AIGS cores experimentally obtained from ICP-AES and TEM (**Supplementary Fig. 14**) are taken into account for the exciton energy calculation.

(Page 19, Supplementary References section)

1. Cragg, G. E. & Efros, A. L. Suppression of Auger processes in confined structures. *Nano Lett.* **2010**, 10(issue), 313-317.
2. Park, K. & Weiss, S. Design rules for membrane-embedded voltage-sensing nanoparticles. *Biophysical journal* **2017**, 112(issue), 703-713.

Reviewer #4 (Publish after major revision):

General Comment: Authors report the heteroepitaxy for AIGS-AgGaS₂ (AIGS₃₂-AGS) core-shell NCs with near-unity PL QYs in the visible range (460 nm to 620 nm) and their optical/structural properties as well as application in displays and luminescent solar concentrators. The AIGS NCs exhibit the best luminescent performances such as high PL QY, high absorption cross-section and narrow emission linewidth than the NCs reported previously. The experimental results are novel and very interesting. Therefore, this manuscript is recommended for publication in *Nature Communications* after the following issues are considered carefully.

Response: We express deep appreciation to Reviewer #4's evaluation (**Publish after revision noted**), valuable feedback and constructive comments on our work. All authors thoroughly went over all comments and suggestions, prepared the responses, and revised the manuscript accordingly. Specifically, per Reviewer #4's comment, we have substantially revised the manuscript and Supplementary Information to detail the synthetic procedures and experimental results. Below, we enclose the point-by-point responses to each comment raised by the Reviewer #4. We look forward that the revised version of manuscript meets high standards of *Nature Communications*.

Comment #4-1: Since high performance AIGS-AGS core-shell NCs were synthesized in this work, authors need to discuss the growth processes of NCs in detail. Especially they should provide clear description and supporting information. Authors think that the key to synthesize high performance AIGS-AGS NCs is the use of Ag-S-Ga(OA)₂ complex. The growth mechanism of NCs is only supported by MALDI-TOF. Nuclear magnetic spectroscopy may provide more information on growth kinetics of NCs.

Response: As the reviewer suggested, in-depth understand of the growth mechanism could offer effective means to control the key structural features of NCs, such as the size, composition, and surface, and thus their photophysical characteristics. In the present study, we have synthesized AIGS-AGS NCs in a batch reactor by sequentially injecting chemical precursors containing Ag, In, Ga, S and organic ligands (plus halide ions) bound to these chemicals. The chemical intermediates are air-sensitive and easily precipitated at room temperature, and thus the separation of these intermediates into individual chemical species to conduct NMR spectroscopy was not allowed. We would like to point out that only few studies have reported NMR spectroscopy to probe NC structures¹¹⁻¹⁴, which are mostly limited to the analysis of ligands that are bound to the surface of NCs.

Instead, we have conducted comparative study to understand the growth mechanism of AIGS NCs. Specifically, we have varied the chemical structures of precursors, their amounts, and their injection sequences, and evaluated the photophysical characteristics together with structural analysis (TEM or XRD) of resulting NCs. After a long and tedious process, we have learned that balancing the reactivity of cation precursors (Ag, In, and Ga) is the key for attaining homogenous AIGS NCs, and S and alkyl amine play a critical role for attaining homogeneous nucleation of AIGS cores and uniform growth of AGS shell. From comparative experiments, we could define the reaction scheme for AIGS NC growth with key intermediates. First, S and alkyl amine reacts to produce in-situ H₂S (which is corrosive and highly reactive) that reacts with Ag₂S to form AgSH. Second, AgSH attacks Ga(OA)₃ to form Ag-S-Ga(OA)₂. Third, Ag-S-Ga(OA)₂ reacts with In(OA)₃ and S to promote nucleation and subsequent growth of homogenous AIGS NCs. Each reaction step and key elements are verified by UV-Vis spectroscopy, chemical composition analysis, MALDI-TOF mass spectroscopy and TEM, which are provided in the original manuscript (**Page 5, lines 94 – 128, Page 15, line 308-339** and **Fig. 1a-b**) and Supplementary Information (**Supplementary Fig. 2-6**).

To aid readers' understanding, we have detailed the synthetic procedures for AIGS-AGS NCs in the revised version of manuscript and Supplementary Information as below.

Change made in the manuscript

(Page 5, line 103 – 130)

The role of in-situ generated H₂S in preparation of Ag-S-Ga(OA)₂ is validated by the comparative experiments with *versus* without degassing for H₂S removal (**Supplementary Fig. 3** and **Fig. 4**). Specifically, we prepare S-OAm stock solutions degassed at two different temperatures, *i.e.*, room temperature (RT) and 130 °C (at which, the in-situ generated H₂S gas is degassed and collected in cold trap connected to the Schlenk line as yellowish-brown color) (**Supplementary Fig. 3** and **Fig. 4**). These S-OAm stock solutions (degassed at RT *versus* 130 °C) are separately added into the reaction flasks containing Ag₂S NPs and the reaction temperature is gradually elevated to 210 °C. Ag₂S NPs mixed with S-OAm degassed at RT are decomposed into a form of AgSH (Ag₂S NPs + H₂S → AgSH, molar ratio of Ag:S = 1:1 is confirmed by ICP-AES), as seen in the gradual decrease in absorbance of Ag₂S NPs and TEM analysis (**Supplementary Fig. 3a** and **Fig. 5**). In presence of Ga(OA)₃, AgSH simultaneously reacts with Ga(OA)₃ to yield Ag-S-Ga(OA)₂ complex. By contrast, Ag₂S NPs mixed with S-OAm degassed at 130 °C remain unchanged until 170 °C, and decompose into elemental Ag and S at 210 °C (**Supplementary Fig. 3b**). In this case, the present Ga(OA)₃ does not participate in the reaction (**Supplementary Fig. 4**). Above results clearly show that in-situ generated H₂S is the key element to produce Ag-S-Ga(OA)₂ complex.

The formation of Ag-S-Ga(OA)₂ is confirmed with MALDI-TOF mass spectrometry and optical characterization (**Fig. 1b, Supplementary Fig. 5** and **Fig. 6**). Ag-S-Ga(OA)₂ stock solution is kept at the room temperature before it was used for AIGS core synthesis and AGS shell growth. The injection of In precursor (In(OA)₃) into Ag-S-Ga(OA)₂ stock solution at an elevated temperature ($T = 210$ °C) bursts AIGS nucleation and subsequent growth. The addition of Ag-S-Ga(OA)₂ stock solution in the AIGS core

containing solution allows to grow AGS shell on top of AIGS core. HR-TEM, STEM-HAADF and EDS analysis verify that resulting NCs are indeed heterostructured in AIGS-AGS core-shell geometry (Fig. 1c-e). It is noted that excess amount of S is deployed throughout the core-shell growth as the S-rich reaction condition guarantees higher PL QY and suppressed trap emission of resulting AIGS-AGS NCs (Supplementary Fig. 7). Reaction procedures for precursor preparations and core-shell growth are detailed in Methods and Supplementary Note 1.

(Page 15, line 312-345)

Ag-S-Ga(OA)₂ preparation. All synthesis was carried out under N₂ atmosphere through the Schlenk line technique. For preparing Ag₂S nanoparticles (NPs), 1 mmol of silver acetate and 5 mL of oleylamine (OAm) were loaded in a three-neck flask and degassed at 50 °C for 1 hr. After the flask was backfilled with N₂, 2.5 ml of dodecanethiol (DDT) and 1 ml of 0.5 M S dissolved in OAm (S-OAm) were injected into the reaction flask to form Ag₂S NPs. To transform Ag₂S NPs to Ag-S-Ga(OA)₂, 4.5 ml of 0.5 M Ga(OA)₃ and 8 ml of 0.5 M S-OAm were added into the reaction flask and temperature was elevated to 210 °C. We note that excess Ga(OA)₃ and S were deployed for complete conversion of Ag₂S NPs to Ag-S-Ga(OA)₂, and the unreacted Ga(OA)₃ and S were used for AIGS core synthesis.

AIGS core synthesis. Reaction flask containing 1.5 ml of Ag-S-Ga(OA)₂ stock solution and 5 ml of OAm is filled with N₂ and heated up to 210 °C. 0.15 ml of 0.5 M In(OA)₃ precursor were swiftly injected into the flask for the nucleation and subsequent growth of AIGS cores (X = 0.5, r = 1.8 nm). We note that an addition of extra S precursor is not needed because excess S-OAm in Ag-S-Ga(OA)₂ stock solution spontaneously participates in the reaction. The reaction temperature was maintained at the elevated temperature for 30 min and cooled to room temperature to cease the reaction. The injected volume of In(OA)₃ precursor was controlled to vary the In content (X) in AIGS cores. Specifically, 0.05, 0.3, 0.6 and 2.7 ml of 0.5 M In(OA)₃ precursor yielded AIGS NCs with In contents (X) of 0.2, 0.6, 0.7 and 0.9, respectively. Resulting AIGS cores were purified twice in glovebox by precipitation (ethanol)/redispersion (toluene) method and finally dispersed in 5 ml of toluene.

AGS shell growth. A reaction flask containing 5 ml of OAm and 300 mg of AIGS cores (X = 0.5) was degassed at 110 °C for 1 hr, back-filled with N₂ and heated up to 240 °C for AGS shell growth. At the elevated temperature, 0.3 ml of Ag-S-Ga(OA)₂ precursor solution, and 0.15 ml of 0.5 M S-OAm were injected, and the reaction temperature was maintained for 1 hr to grow 0.3 nm thick AGS shell. Alternatively, equivalent amount of Ag₂S NPs, Ga(OA)₃ and S-OAm solution were added for AGS shell growth. It is noted that when shell thickness exceeded 0.6 nm, 10 M GaCl₃ solution (dissolved in ethanol, 1 eq of Ga(OA)₃) was added to facilitate uniform shell growth³⁹. For thicker AGS shell growth, Ag, Ga and S precursors were injected repeatedly to grow AGS shell (0.3-0.5 nm for each step) at the fixed reaction temperature (240 °C). Excess amount of S compared to Ag or Ga is deployed throughout the reaction for higher PL QY and suppressed trap emission of AIGS-AGS NCs. Resulting AIGS-AGS NCs were purified twice in glovebox by precipitation (ethanol)/redispersion (toluene) method and finally dispersed in 5 ml of toluene for further characterization and applications.

Change made in Supplementary Information
(Supplementary Fig. 6)

Supplementary Fig. 6 | MALDI-TOF mass spectra of (a) Ga(OA)₃ and (b) Ag-S-Ga(OA)₂. Dashed red lines in (a) and (b) represent theoretical m/z value from Ag and Ga isotopes. The slight deviation from theoretical m/z value of Ag-S-Ga(OA)₂ is likely originated from Ga(OA)₃, whose measured values also deviate from theoretical value.

(Supplementary Fig. 7)

Supplementary Fig. 7 | Absorption and PL spectra of AIGS ($X = 0.5$, $r = 1.8$ nm)-AGS ($l = 1.6$ nm) NCs synthesized from S rich reaction condition (Ag:Ga:S = 1:4.5:10, black line) versus Ga rich reaction condition (Ag:Ga:S = 1:4.5:4.5, red line). The inset shows the magnified view of tail emission in PL spectra.

(Supplementary Fig. 9)

Supplementary Fig. 9 | (a-d) STEM-HAADF image and EDS elemental mapping of Ag (red), In (green), Ga (blue) and S (yellow) for AIGS ($X = 0.5$, $r = 1.8$ nm)-AGS ($l = 1.2$ nm) NCs. (e) Composition ratios of individual AIGS ($X = 0.5$, $r = 1.8$ nm)-AGS ($l = 1.2$ nm) NC marked in Supplementary Fig. 9a. Scale bars are 5 nm.

(Supplementary Fig. 11)

Supplementary Fig. 11 | (a) Absorption, PL spectra and (b) normalized PL spectra of AIGS ($X = 0.5$, $r = 1.8 \text{ nm}$)-AGS ($l = 1.6 \text{ nm}$) NCs as a function of purification steps (precipitation with ethanol and redispersion with toluene). The inset of (b) shows PL QY of AIGS ($X = 0.5$, $r = 1.8 \text{ nm}$)-AGS ($l = 1.6 \text{ nm}$) NCs as a function of purification steps. Addition of TOP into 5-times purified AIGS-AGS NC solution aids to recover PL QY to its initial value.

Comment #4-2: The description of the picture information is not exhaustive. The information shown in the picture is not reflected in the text description.

Response: We noticed that the image in the **Fig. 2g** was not adequately explained in the previous version of the manuscript. We revised our manuscript for clear description of the image. All other pictures in the manuscript are described properly such as the image in **Fig. 4d**, which was explained in the caption as schematic and photographic image of a luminescent solar concentrator (LSC), whose size is 100 x 50 x 10 mm³. The image in the **Supplementary Fig. 2** was described both in the Figure and the caption as Photographic images of silver acetate (Agac) solution (a) at 50 °C and (b) 170 °C. Black precipitates are reduced Ag. Similarly, the image in **Supplementary Fig. 5** is also properly described in the caption as photographic images of (d) Ag₂S NPs, (e) Ag-S-Ga(OA)₂ complex, and (f) AIGS ($X = 0.5$, $r = 1.8$ nm) cores.

Change made in the manuscript

(Page 27, Caption of Fig. 2g)

Fig. 2 | AIGS-AGS NCs with variable core compositions and shell dimensions. (a) Schematic illustrations of the geometry (left) and potential profile (right) of AIGS-AGS NCs. Blue and red lines indicate the lowest quantized energy state for electron (1S_c) and hole (1S_v), respectively. (b) TEM images (scale bars = 10 nm), (c) X-ray diffraction patterns, and (d) UV-Vis absorption and PL spectra of AIGS ($X = 0.5$, $r =$

1.8 nm)-AGS NCs with varying shell thicknesses ($0 \leq l \leq 2.9$ nm). The inset displays PL QYs as a function of shell thickness. 2d plots of PL spectra are supported in **Supplementary Fig. 8** (e) PL spectra, (f) energy gaps ($1S_e-1S_h$) obtained from UV-Vis spectra (top) and PL QYs (bottom), and (g) their color coordinates of AIGS-AGS ($l = 1.6$ nm) NCs with varying In contents ($0.2 \leq X \leq 0.9$) (inset: a photographic image of AIGS-AGS NCs with varying In ratios ($X = 0.2, 0.3, 0.5, 0.7$ and 0.9 from the left)). Gray triangles in (f) represent our calculation results for which homogeneously alloyed AIGS cores are taken into account. Synthesis and experimental results for AIGS-AGS NCs with varying compositions and shell dimensions are detailed in **Methods, Supplementary Note 1** and **Supplementary Fig. 7-16**.

Comment #4-3: Why are the NCs with core radius $r=1.8$ nm used by the author in Fig. 2, while the NCs with core radius $r=1.6$ nm are used to describe the optical properties in Fig. 3? How to select NCs with the $r=1.8$ nm and 1.6 nm? How does the size of NC core affect the optical properties of quantum dots?

Response: We thank Reviewer #3 for pointing out our mistake. The radius within in **Fig. 3** is miswritten. The mean radius is 1.8 nm for all NCs in **Fig. 3**. We fixed the figure legend in **Fig. 3e**. In addition, we thoroughly went over all other figures to make sure their legends are given correctly.

Change made in the manuscript
(Page 28, legends in Fig. 3e)

Fig. 3 | Impact of AGS heteroepitaxy on photophysical and photochemical properties of individual AIGS-AGS NCs. (a) Representative PL intensity trajectories (left) and PL intensity histograms (right) of AIGS-AGS NC (top, cyan) versus AIGS core (bottom, gray) (bin time = 50 ms). (b) On-time fraction statistics of individual AIGS-AGS NCs (left, cyan) and AIGS cores (right, gray). Average on-times are 63.9 % and 12.3 % for AIGS-AGS NCs and AIGS cores, respectively. (c) Emission spectra from $1S_e-1S_h$ transition (bottom, AIGS-AGS NC) and surface trap states (top, AIGS core). The composition and dimensions are $X = 0.5$, $r = 1.8$ nm, and $l = 0$ or 1.6 nm for AIGS cores or AIGS-AGS NCs, respectively. The insets depict luminescence mechanism and its characteristic time. (d) PL peak versus FWHM distribution in individual AIGS-AGS NCs or AIGS cores. (e) Time dependent PL QYs of AIGS ($X = 0.5$, $r = 1.8$ nm)-AGS NCs with varying shell thicknesses ($l = 0.6, 1.2, 1.6$ or 2.9 nm) upon oxidative test. The inset displays T_{50} , the time when PL QY reaches to the half of initial values. See **Supplementary Fig. 17** for detailed optical characteristics (exciton lifetimes and $g^{(2)}(t)$) of individual NCs.

Comment #4-4: The component description of quantum dots is not accurate enough. The authors use In content (X) in AIGS core to describe components. X is needed to be described or defined, for example, $X = \text{In}/\text{Ga}$ or $X = \text{In}/(\text{In} + \text{Ga})$ or $X = \text{In}/(\text{Ag} + \text{In} + \text{Ga} + \text{S})$.

Response: X implies $[\text{In}]/([\text{In}] + [\text{Ga}])$ throughout the manuscript. We clarified the definition of X in the revised version of manuscript.

Change made in the manuscript

(Page 7, line 151)

To obtain In content (X) in AIGS cores, we carry out elemental analysis (ICP-AES) or estimate it from XRD. Here, X is $[\text{In}]/([\text{In}] + [\text{Ga}])$.

Comment #4-5: When authors study the effect of components on PLQY of NCs, they obtain the PL QY of 50% for the NCs with $X=0.2$. The reason of low PLQY is related to the large band gap of cores and the AGS shell coating that cannot effectively limit the carriers. It is noted that in **Fig. 2f** the overall PLQY also shows a downward trend with the increase of X when $X>0.5$. This obviously cannot be explained by the model of shell confined carriers.

Response. PL QYs of AIGS-AGS NCs appear to decrease with increasing the In content (X) in AIGS cores (**Fig. 2f**) and Reviewer #4 poses a question on the origin for the trend. We would like to underline the fact that, despite the appearing decrease, AIGS-AGS NCs with higher In contents ($X > 0.5$) still show narrow gaussian shape PL with PL QYs greater than 85 %. These photophysical characteristics imply the effective recombination of photoexcited charge carriers into photons in chosen sample variations.

We attribute the reason for relatively lower PL QYs for AIGS-AGS NCs with higher In contents to un-optimized synthetic conditions for them. In fact, the synthetic condition for green-emitting AIGS-AGS NCs ($X = 0.5$) is meticulously optimized in response to the industrial call for the replacement of commercialized green-emitting InP QDs to ones having higher extinction coefficient and narrower spectral linewidth. As a result of careful optimization, green-emitting AIGS-AGS NCs ($X = 0.5$) have been improved to show near-unity PL QYs with FWHM of ~ 30 nm. We believe that further optimization of the reaction conditions for uniform growth of shells and effective passivation of surface trap states can also improve PL QYs of AIGS-AGS NCs with higher In contents ($X > 0.5$) comparable to the green ones ($X = 0.5$).

Comment #4-6: Most of the reported narrow-band AIGS quantum dots have a long tail in PL bands. In this article, **Fig. 2e** and **Fig. S16** show different degrees of tailing in PL bands only when $X=0.2$ and $X=0.9$. Authors should highlight this point and give reasonable explanations.

Response: In core-shell NC having a quantum-well like potential profile, the photoexcited charge carriers that are effectively confined within the core recombine to yield photons having energies corresponding to the energy gap between lowest quantized states for electron and hole. In an ideal case (without the presence of interfacial or surface defects), core-shell NCs with the straddling gap show narrow-band gaussian shape PL spectra with near-unity single exciton PL QYs. AIGS ($X = 0.5$)-AGS NC displaying narrow-band PL emission with near-unity PL QY is the example of the ideal case.

In a non-ideal case in which interfacial or surface defects present, the radiative recombination process of charge carriers should compete with charge trapping to defects that accompanies trap-stated mediated emissions or non-radiative Auger recombination processes upon continued photoexcitation. The presence of interfacial or surface defects in core-shell NCs comes with the rise of broad-band trap emission and/or the decrease in PL QYs. AIGS ($X = 0.2$)-AGS NCs show a long tail in PL spectra 300 meV apart from the main peak and PL QY of $\sim 50\%$, signifying the trapping of photoexcited charge carriers to either interfacial or surface trap states. Considering the small lattice mismatch between AIGS core and AGS shell (even smaller than the case with AIGS ($X = 0.5$)-AGS NCs), we speculate the creation of interfacial defects is unlikely and thus trapping of photoexcited charge carriers to the surface states is the main culprit. Assuming that all AIGS ($X = 0.2$)-AGS NCs have similar surface properties with these of AIGS ($X = 0.5$)-AGS NCs, the promoted surface trapping of photoexcited charge is attributed to the larger energy gap of AIGS core ($X = 0.2$) compared to AGS shell ($\Delta E = 100$ meV for electron and $\Delta E = 50$ meV for hole).

We consider the case with AIGS ($X = 0.9$)-AGS NCs is different from the case with $X = 0.2$. In fact, AIGS ($X = 0.9$)-AGS NCs have single gaussian shape PL spectra with PL QYs greater than 85%, implying the efficient recombination of charge carriers in AIGS cores. The comparison of intensity-normalized, peak centered PL spectra of AIGS-AGS NCs with varying In contents ($0.2 \leq X \leq 0.9$) clearly shows that AIGS ($X = 0.9$)-AGS NCs does not entail a tail emission at the lower energy regime (**Fig. R10**).

Fig. R10 | PL spectra of AIGS-AGS ($l = 1.6$ nm) NCs with varying In contents ($0.2 \leq X \leq 0.9$) presented in **Fig. 2e**. each spectrum are centered at respective PL Peak.

Change made in the manuscript

(Page 27, range of x-axis in Fig. 2e)

Fig. 2 | AIGS-AGS NCs with variable core compositions and shell dimensions. (a) Schematic illustrations of the geometry (left) and potential profile (right) of AIGS-AGS NCs. Blue and red lines indicate the lowest quantized energy state for electron ($1S_e$) and hole ($1S_h$), respectively. (b) TEM images (scale bars = 10 nm), (c) X-ray diffraction patterns, and (d) UV-Vis absorption and PL spectra of AIGS ($X = 0.5$, $r = 1.8$ nm)-AGS NCs with varying shell thicknesses ($0 \leq l \leq 2.9$ nm). The inset displays PL QYs as a function of shell thickness. 2d plots of PL spectra are supported in **Supplementary Fig. 8** (e) PL spectra, (f) energy gaps ($1S_e - 1S_h$) obtained from UV-Vis spectra (top) and PL QYs (bottom), and (g) their color coordinates of AIGS-AGS ($l = 1.6$ nm) NCs with varying In contents ($0.2 \leq X \leq 0.9$) (inset: a photographic image of AIGS-AGS NCs with varying In ratios ($X = 0.2, 0.3, 0.5, 0.7$ and 0.9 from the left)). Gray triangles in (f) represent our calculation results for which homogeneously alloyed AIGS cores are taken into account. Synthesis and experimental results for AIGS-AGS NCs with varying compositions and shell dimensions are detailed in **Methods**, **Supplementary Note 1** and **Supplementary Fig. 7-16**.

Comment #4-7: Can the optical properties of AIGS-AGS NCs be improved further if the ZnS shell with a wider gap will continue to be coated?

Response: Coherent heteroepitaxy with extra inorganic shells having a large bandgap and chemical robustness can enhance photochemical stability of AIGS-AGS NCs. ZnS, which has a large bandgap (3.54 eV) and proven chemical/structural robustness, is of particular interest as the exterior shell, as witnessed in the case of the twin ternary compound CuInS₂-ZnS NCs. However, the growth of ZnS on AIGS without creating defects has not been made¹⁵. **Fig. R11** shows our experimental results, showing spectral linewidth broadening (fwhm 36 nm to 50 nm) together with PL QY drop from 93 % to 50 % when the zinc precursor (*i.e.*, Zn(OA)₂) is introduced into AIGS-AGS NC solutions. This indicates that the ion exchange occurs among Zn and cations (*i.e.*, Ag, Ga) and creates trap states. Therefore, we consider that the improvement of optical properties of AIGS-AGS NCs with ZnS exterior shell is unlikely, unless new chemistry for defect-free heteroepitaxy of ZnS on AGS or AIGS is developed.

Fig. R11 | Absorption and PL spectra of AIGS ($X = 0.5$, $r = 1.8$ nm)-AGS ($l = 1.2$ nm) before *versus* after Zn(OA)₂ incorporation.

Comment #4-8: In Fig. 2C, the XRD diffraction pattern of AIGS cores appears to deviate from the standard Tetragonal-AgInS₂, which should be explained.

Response: AIGS cores are alloyed compounds of AgInS₂ and AgGaS₂, and thus their XRD patterns should deviate either from standard tetragonal AgInS₂ or AgGaS₂. The extent of deviation in XRD patterns of AIGS cores from AgInS₂ becomes greater for AIGS having greater Ga contents (**Supplementary Fig. 13f**). We acquire the (In,Ga) contents from XRD patterns and confirmed these values coincide with ICP analysis (**Supplementary Fig. 14**).

Comment #4-9: In Fig. S17, (b,d) should be (b,e) in the caption.

Response: We thank Reviewer #4 for pointing out a typo in the manuscript. We reflected Reviewer #4's comment in the revised version of Supplementary Information.

***Change made in Supplementary Information
(Caption of Supplementary Fig. 17e)***

Supplementary Fig. 17 | Individual NC photophysical characteristics of AIGS ($X = 0.5$, $r = 1.8$ nm)-AGS ($l = 1.6$ nm) NC and AIGS core ($X = 0.5$, $r = 1.8$ nm) (a,d) PL spectrum, (b,e) photon-photon correlation function with $g^{(2)}(0) = 0.014$ and 0.005 , respectively, and (c,f) PL decay curves.

References

1. Whitham, P. J. et al. Single-particle photoluminescence spectra, blinking, and delayed luminescence of colloidal CuInS₂ nanocrystals. *J. Phys. Chem. C* **120**, 17136-17142 (2016).
2. Zang, H. et al. Thick-shell CuInS₂/ZnS quantum dots with suppressed “blinking” and narrow single-particle emission line widths. *Nano Lett.* **17**, 1787-1795 (2017).
3. Gungor, K., Du, J. & Klimov, V. I. General Trends in the Performance of Quantum Dot Luminescent Solar Concentrators (LSCs) Revealed Using the “Effective LSC Quality Factor”. *ACS Energy Letters* **7**, 1741-1749 (2022).
4. Klimov, V. I., Baker, T. A., Lim, J., Velizhanin, K. A. & McDaniel, H. Quality factor of luminescent solar concentrators and practical concentration limits attainable with semiconductor quantum dots. *Acs Photonics* **3**, 1138-1148 (2016).
5. Bergren, M. R. et al. High-Performance CuInS₂ Quantum Dot Laminated Glass Luminescent Solar Concentrators for Windows. *ACS Energy Letters* **3**, 520-525 (2018).
6. Velarde, A. s. R. et al. Optimizing the aesthetics of high-performance CuInS₂/ZnS quantum dot luminescent solar concentrator windows. *ACS Applied Energy Materials* **3**, 8159-8163 (2020).
7. Liu, X. et al. Eco-friendly quantum dots for liquid luminescent solar concentrators. *Journal of Materials Chemistry A* **8**, 1787-1798 (2020).
8. Carlos, C. P. et al. Environmentally friendly luminescent solar concentrators based on an optically efficient and stable green fluorescent protein. *Green Chemistry* **22**, 4943-4951 (2020).
9. Krumer, Z., van Sark, W. G., Schropp, R. E. & de Mello Donegá, C. Compensation of self-absorption losses in luminescent solar concentrators by increasing luminophore concentration. *Solar Energy Materials and Solar Cells* **167**, 133-139 (2017).
10. Cragg, G. E. & Efros, A. L. Suppression of Auger processes in confined structures. *Nano Lett.* **10**, 313-317 (2010).
11. Fritzing, B., Capek, R. K., Lambert, K., Martins, J. C. & Hens, Z. Utilizing self-exchange to address the binding of carboxylic acid ligands to CdSe quantum dots. *J. Am. Chem. Soc.* **132**, 10195-10201 (2010).
12. Fritzing, B. et al. In situ observation of rapid ligand exchange in colloidal nanocrystal suspensions using transfer NOE nuclear magnetic resonance spectroscopy. *J. Am. Chem. Soc.* **131**, 3024-3032 (2009).
13. Hassinen, A., Moreels, I., de Mello Donegá, C., Martins, J. C. & Hens, Z. Nuclear magnetic resonance spectroscopy demonstrating dynamic stabilization of CdSe quantum dots by alkylamines. *J. Phys. Chem. Lett.* **1**, 2577-2581 (2010).
14. Choi, Y., Hahm, D., Bae, W. K. & Lim, J. Heteroepitaxial chemistry of zinc chalcogenides on InP nanocrystals for defect-free interfaces with atomic uniformity. *Nat. Commun.* **14**, 43 (2023).
15. Kim, J.-H. et al. Synthesis of widely emission-tunable Ag–Ga–S and its quaternary derivative quantum dots. *Chem. Eng. J.* **347**, 791-797 (2018).

REVIEWER COMMENTS

Reviewer #1 (Remarks to the Author):

The authors have taken my comments seriously and went a long way to provide satisfactory to all of them.

However, in the attempt to reply to my third comment (comparison with state of the art LSCs), the authors performed some calculations that might be misleading. In particular, the Monte Carlo simulation performed with hypothetical QDs emitting at 1000nm. I understand the rationale behind such a choice but the solid shift of the PL spectrum to 1000 nm provides an incorrect estimate of the spectral overlap with the respective absorbance and hence of the LSC efficiency. This is because the absorption spectrum retains its shape when solidly shifted (it is barely the ratio between incident and transmitted light) but the PL peak broadens in the IR because of its basic definition (The PL is the integral in $d\lambda$; in other words the FWHM stays hypothetically constant in energy but it increases in the NIR when evaluated in wavelength). Solidly shifting the spectrum theoretically implies that the profile in the visible is substantially narrower than the actual experimental one.

I strongly suggest to remove the last part of the calculation.

Also, the comment on the impossibility to synthesize CIS or AIS QDs emitting in the visible range is a stretch as it is well known that both compounds alloyed with Zinc retain most of their optical properties and emit in the visible if properly size tuned.

Overall the paper is more solid than the original version and could be suitable for NComm once these last aspects have been addressed.

Reviewer #3 (Remarks to the Author):

While the issue of MALDI-MS measurements deviating from theoretical values remains unresolved, it is addressed in the supplementary figure. In other respects, the paper is generally well revised and deserves publication in Nature Communications. However, it seems that references to Supplementary Figures 8, 9, and 11 are missing in the main text. Please check the submission guidelines to see if it is acceptable to just show the figures without referencing them in the text.

Reviewer #4 (Remarks to the Author):

This manuscript has been well revised. Therefore it is recommended for publication in Nature Communications.

Response to Reviewers

Reviewer #1 (Publish after minor revision noted):

General Comment: The authors have taken my comments seriously and went a long way to provide satisfactory to all of them. However, in the attempt to reply to my third comment (comparison with state-of-the-art LSCs), the authors performed some calculations that might be misleading. In particular, the Monte Carlo simulation performed with hypothetical QDs emitting at 1000nm. I understand the rationale behind such a choice but the solid shift of the PL spectrum to 1000 nm provides an incorrect estimate of the spectral overlap with the respective absorbance and hence of the LSC efficiency. This is because the absorption spectrum retains its shape when solidly shifted (it is barely the ratio between incident and transmitted light) but the PL peak broadens in the IR because of its basic definition (The PL is the integral in $d\lambda$; in other words the FWHM stays hypothetically constant in energy but it increases in the NIR when evaluated in wavelength). Solidly shifting the spectrum theoretically implies that the profile in the visible is substantially narrower than the actual experimental one. I strongly suggest to remove the last part of the calculation.

Also, the comment on the impossibility to synthesize CIS or AIS QDs emitting in the visible range is a stretch as it is well known that both compounds alloyed with Zinc retain most of their optical properties and emit in the visible if properly size tuned.

Overall, the paper is more solid than the original version and could be suitable for NComm once these last aspects have been addressed.

Response: We express our sincere gratitude to Reviewer #1 for his/her constructive comments and suggestions on our work.

Per Reviewer #1's concern on the validity of our calculation, in which ultra-narrow PL spectral linewidth is presumed for AIGS NCs with NIR bandgap, we concede the reviewer's point that the calculation results could notably deviate from the experimental results. To avoid unwanted misleading, we have decided not to include the calculation results either in the manuscript or Supplementary Information. (In fact, the calculation results were provided only in the previous Response Letter, and thus the Manuscript or Supplementary Information remains unchanged.)

Per Reviewer #1' comment on the emission energy range for CIS NCs, we have specified experimentally achievable energy range of CIS NCs in the revised version of Manuscript.

Change made in the manuscript (Page 14, line 290-302)

We conclude by noting that the photophysical properties of AIGS-AGS NCs are in stark contrast to those of its twin compound CuInS_2 (CIS) NCs, thereby expanding the application scope of I-III-VI₂ NCs. CIS NCs exhibit high PL QYs in the green to near-IR region (peak PL energy from 1.5 eV to 2.4 eV) with large Stokes shift (300-500 meV), broad emission linewidth (FWHM > 400 meV), and delayed radiative decay time (~200 ns) due to Cu hole trap states^{44,45} from two different oxidation states of Cu, namely cuprous (Cu^+) and cupric (Cu^{2+}). These characteristics guarantee a wide range of solar spectrum absorption and the suppression of energy loss *via* reabsorption processes, which are best adapted for high efficiency LSCs^{20,46-49}. AIGS-AGS NCs, in contrast, exhibit high PL QYs in the visible region (peak PL energy from 2.0 eV to 2.6 eV) with narrow emission linewidth (FWHM ~ 140 meV) and rapid radiative decay time (~40 ns), making them suitable for use in displays, lightings, and colorful LSCs^{24,36-39}. In particular, the LSC with AIGS-AGS NCS will provide aesthetic appealing to energy harvesting system incorporating architectures with small losses^{22,50-52}.

Reviewer #3 (Publish after minor revision noted):

General Comment: While the issue of MALDI-MS measurements deviating from theoretical values remains unresolved, it is addressed in the supplementary figure. In other respects, the paper is generally well revised and deserves publication in Nature Communications. However, it seems that references to Supplementary Figures 8, 9, and 11 are missing in the main text. Please check the submission guidelines to see if it is acceptable to just show the figures without referencing them in the text.

Response: We sincerely appreciate Reviewer #3's positive evaluation on our work. Per reviewer's suggest, we have referenced **Supplementary Figures 8,9 and 11** in the revised Manuscript.

Change made in the manuscript

(Page 16, line 346-347)

AGS shell growth. A reaction flask containing 5 ml of OAm and 300 mg of AIGS cores ($X = 0.5$) was degassed at 110 °C for 1 hr, back-filled with N_2 and heated up to 240 °C for AGS shell growth. At the elevated temperature, 0.3 ml of Ag-S-Ga(OA)₂ precursor solution and 0.15 ml of 0.5 M S-OAm were injected, and the reaction temperature was maintained for 1 hr to grow 0.3 nm thick AGS shell. Alternatively, equivalent amount of Ag₂S NPs, Ga(OA)₃ and S-OAm solution were added for AGS shell growth. It is noted that when shell thickness exceeded 0.6 nm, 10 M GaCl₃ solution (dissolved in ethanol, 1 eq of Ga(OA)₃) was added to facilitate uniform shell growth³⁹. For thicker AGS shell growth, Ag, Ga and S precursors were injected repeatedly to grow AGS shell (0.3-0.5 nm for each step) at the fixed reaction temperature (240 °C). Excess amount of S compared to Ag or Ga is deployed throughout the reaction for higher PL QY and suppressed trap emission of AIGS-AGS NCs. Resulting AIGS-AGS NCs were purified twice in glovebox by precipitation (ethanol)/redispersion (toluene) method and finally dispersed in 5 ml of toluene for further characterization and applications. **Changes in the optical properties of NCs during the repeated purification steps are provided in Supplementary Fig. 11.**

Fig. 1 | Ag(In,Ga)₂-AgGaS₂ (AIGS-AGS) core-shell NCs. (a) Schematic illustration of AIGS-AGS NC synthesis. Key reactants and chemical intermediates are depicted. (b) MALDI-TOF mass spectra of the reactants containing Ag, S, Ga and ligands (oleic acid (OA) and oleyl amine (OAm)) taken at different reaction temperatures (130 °C (top, cyan) and 50 °C (bottom, gray)). The colored background highlights the presence of Ag-S-Ga(OA)₂ complex at the elevated temperature. The spectra are vertically shifted for visual clarity. (c,d) HR-TEM images of AIGS ($X = 0.5$, radius (r) = 1.8 nm)-AGS (shell thickness (l) = 2.9 nm) NCs. A NC is measured along the [110] axis. (e) STEM-HAADF image and EDS elemental mapping of Ag (red), In (green), Ga (blue) and S (yellow) for AIGS ($X = 0.9$, $r = 2.95$ nm)-AGS ($l = 2.7$ nm) NCs. Scale bars are 5 nm. EDS analysis showing the compositional homogeneity among multiple individual AIGS-AGS NCs is presented in Supplementary Fig. 8.

Fig. 2 | AIGS-AGS NCs with variable core compositions and shell dimensions. (a) Schematic illustrations of the geometry (left) and potential profile (right) of AIGS-AGS NCs. Blue and red lines indicate the lowest quantized energy state for electron ($1S_e$) and hole ($1S_h$), respectively. (b) TEM images (scale bars = 10 nm), (c) X-ray diffraction patterns, and (d) UV-Vis absorption and PL spectra of AIGS ($X = 0.5$, $r = 1.8$ nm)-AGS NCs with varying shell thicknesses ($0 \leq l \leq 2.9$ nm). The inset displays PL QYs as a function of shell thickness. **2d plots of PL spectra are supported in Supplementary Fig. 9** (e) PL spectra, (f) energy gaps ($1S_e - 1S_h$) obtained from UV-Vis spectra (top) and PL QYs (bottom), and (g) their color coordinates of AIGS-AGS ($l = 1.6$ nm) NCs with varying In contents ($0.2 \leq X \leq 0.9$) (inset: a photographic image of AIGS-AGS NCs with varying In ratios ($X = 0.2, 0.3, 0.5, 0.7$ and 0.9 from the left)). Gray triangles in (f) represent our calculation results for which homogeneously alloyed AIGS cores are taken into account. **Synthesis and experimental results for AIGS-AGS NCs with varying compositions and shell dimensions are detailed in Methods, Supplementary Note 1 and Supplementary Fig. 7-16.**

REVIEWERS' COMMENTS

Reviewer #1 (Remarks to the Author):

The authors have addressed the remaining issue according to my suggestion. My recommendation is to accept this paper for publication in its current form.

Response to Reviewers

Reviewer #1 (Publish as is):

General Comment: The authors have addressed the remaining issue according to my suggestion. My recommendation is to accept this paper for publication in its current form.

Response: We sincerely appreciate Reviewer #1's positive evaluation on our work, Reviewer #1's comments in the revision process have been invaluable in improving the overall quality and integrity of our manuscript and enhancing the impact of our research.